# Importance Weighted Actor-Critic for Optimal Conservative Offline Reinforcement Learning

**Hanlin Zhu**
EECS, UC Berkeley
hanlinzhu@berkeley.edu

**Paria Rashidinejad**
EECS, UC Berkeley
paria.rashidinejad@berkeley.edu

**Jiantao Jiao**
EECS and Statistics, UC Berkeley
jiantao@berkeley.edu

## Abstract

We propose A-Crab (Actor-Critic Regularized by Average Bellman error), a new practical algorithm for offline reinforcement learning (RL) in complex environments with insufficient data coverage. Our algorithm combines the marginalized importance sampling framework with the actor-critic paradigm, where the critic returns evaluations of the actor (policy) that are pessimistic relative to the offline data and have a small average (importance-weighted) Bellman error. Compared to existing methods, our algorithm simultaneously offers a number of advantages: (1) It achieves the optimal statistical rate of $1/\sqrt{N}$—where $N$ is the size of offline dataset—in converging to the best policy covered in the offline dataset, even when combined with general function approximators. (2) It relies on a weaker *average* notion of policy coverage (compared to the $\ell_\infty$ single-policy concentrability) that exploits the structure of policy visitations. (3) It outperforms the data-collection behavior policy over a wide range of specific hyperparameters. We provide both theoretical analysis and experimental results to validate the effectiveness of our proposed algorithm. The code is available at https://github.com/zhuhl98/ACrab.

## 1 Introduction

Offline reinforcement learning (RL) algorithms aim at learning a good policy based only on historical interaction data. This paradigm allows for leveraging previously-collected data in learning policies while avoiding possibly costly and dangerous trial and errors and finds applications in a wide range of domains from precision medicine (Tang et al., 2022) to robotics (Sinha et al., 2022) to climate (Rolnick et al., 2022). Despite wide applicability, offline RL has yet to achieve success stories akin to those observed in online settings that allow for trials and errors (Mnih et al., 2013; Silver et al., 2016; Ran et al., 2019; Mirhoseini et al., 2020; Oh et al., 2020; Fawzi et al., 2022; Degrave et al., 2022).

Enabling offline RL for complex real-world problems requires developing algorithms that first, handle complex high-dimensional observations and second, have minimal requirements on the data coverage and "best" exploit the information available in data. Powerful function approximators such as deep neural networks are observed to be effective in handling complex environments and deep RL algorithms have been behind the success stories mentioned above. This motivates us to investigate provably optimal offline RL algorithms that can be combined with general function approximators and have minimal requirements on the coverage and size of the offline dataset.

37th Conference on Neural Information Processing Systems (NeurIPS 2023).

In RL theory, the data coverage requirements are often characterized by concentrability definitions (Munos, 2007; Scherrer, 2014). For a policy $\pi$, the ratio of the state-action occupancy distribution $d^\pi$ of $\pi$ to the dataset distribution $\mu$, denoted by $w^\pi = d^\pi/\mu$, is used to define concentrability. The most widely-used definition is $\ell_\infty$ concentrability, defined as the infinite norm of $w^\pi$, i.e., $C^\pi_{\ell_\infty} = \|w^\pi\|_\infty$. Many earlier works on offline RL require all-policy $\ell_\infty$ concentrability (i.e., $C^\pi_{\ell_\infty}$ is bounded for all candidate policy $\pi$) (Scherrer, 2014; Liu et al., 2019a; Chen and Jiang, 2019; Jiang, 2019; Wang et al., 2019; Liao et al., 2020; Zhang et al., 2020a) or stronger assumptions such as a uniform lower bound on $\mu(a|s)$ (Xie and Jiang, 2021). However, such all-policy concentrability assumptions are often violated in practical scenarios, and in most cases, only *partial* dataset coverage is guaranteed.

To deal with partial data coverage, recent works use conservative algorithms, which try to avoid policies not well-covered by the dataset, to learn a good policy with much weaker dataset coverage requirements (Kumar et al., 2020; Jin et al., 2021). In particular, algorithms developed based on the principle of pessimism in the face of uncertainty are shown to find the best *covered* policy (or sometimes they require coverage of the optimal policy) (e.g., Rashidinejad et al. (2021, 2022); Zhan et al. (2022); Chen and Jiang (2022)). However, most of these works use $\ell_\infty$ concentrability to characterize the dataset coverage. This could still be restrictive even if we only require single-policy concentrability, since the $\ell_\infty$ definition characterizes coverage in terms of the worst-case maximum ratio over all states and actions. Other milder variants of the single-policy concentrability coefficient are proposed by Xie et al. (2021); Uehara and Sun (2021) which consider definitions that exploit the structure of the function class to reduce the coverage requirement and involve taking a maximum over the functions in the hypothesis class instead of all states and actions. However, as we show in Section 2.4, when the function class is very expressive, these variants will degenerate to $\ell_\infty$ concentrability. Moreover, previous algorithms requiring milder variants of single-policy concentrability are either computationally intractable (Xie et al., 2021; Uehara and Sun, 2021) or suffer a suboptimal rate of suboptimality (Cheng et al., 2022). Therefore, a natural and important question is raised:

> *Is there a computationally efficient and statistically optimal algorithm that can be combined with general function approximators and have minimal requirements on dataset coverage?*

We answer this question affirmatively by proposing a novel algorithm named A-Crab (Actor-Critic Regularized by Average Bellman error). We also discuss more related works in Appendix A.

Table 1: Comparison of provable offline RL algorithms with general function approximation.

| Algorithm | Computation | Any covered policy | Coverage assumption | Policy improvement | Suboptimality |
|---|---|---|---|---|---|
| Xie et al. (2021) | Intractable | Yes | single-policy, $C^\pi_{\text{Bellman}}$ | Yes | $O\left(\frac{1}{\sqrt{N}}\right)$ |
| Uehara and Sun (2021) | Intractable | Yes | single-policy, $\ell_\infty$ and $\ell_2$ | Yes | $O\left(\frac{1}{\sqrt{N}}\right)$ |
| Chen and Jiang (2022) | Intractable | No | single-policy, $\ell_\infty$ | No | $O\left(\frac{1}{\sqrt{N}\text{gap}(Q^\star)}\right)$ |
| Zhan et al. (2022) | Efficient | Yes | two-policy, $\ell_\infty$ | Yes | $O\left(\frac{1}{N^{1/6}}\right)$ |
| Cheng et al. (2022) | Efficient | Yes | single-policy, $C^\pi_{\text{Bellman}}$ | Yes & Robust | $O\left(\frac{1}{N^{1/3}}\right)$ |
| Rashidinejad et al. (2022) | Efficient | No | single-policy, $\ell_\infty$ | No | $O\left(\frac{1}{\sqrt{N}}\right)$ |
| Ozdaglar et al. (2022) | Efficient | No | single-policy, $\ell_\infty$ | No | $O\left(\frac{1}{\sqrt{N}}\right)$ |
| A-Crab (this work) | Efficient | Yes | single-policy, $\ell_2$ | Yes & Robust | $O\left(\frac{1}{\sqrt{N}}\right)$ |

## 1.1 Contributions

In this paper, we build on the adversarially trained actor-critic (ATAC) algorithm of Cheng et al. (2022) and combine it with the marginalized importance sampling (MIS) framework (Xie et al., 2019; Chen and Jiang, 2022; Rashidinejad et al., 2022). In particular, we replace the squared Euclidean norm of the Bellman-consistency error term in the critic's objective of the ATAC algorithm with an importance-weighted average Bellman error term. We prove that this simple yet critical modification of the ATAC algorithm enjoys the properties highlighted below (see Table 1 for comparisons with previous works).

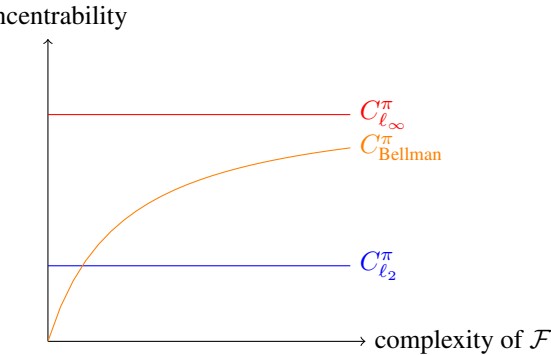

Figure 1: An intuitive comparison of $C_{\ell_\infty}^\pi$, $C_{\ell_2}^\pi$ and $C_{\text{Bellman}}^\pi$ for a fixed policy $\pi$. Note that $C_{\ell_\infty}^\pi$, $C_{\ell_2}^\pi$ are independent of the function class $\mathcal{F}$, and $C_{\ell_2}^\pi$ is always smaller than $C_{\ell_\infty}^\pi$. $C_{\text{Bellman}}^\pi$ grows as $\mathcal{F}$ gets richer and converges to $C_{\ell_\infty}^\pi$.

**1. Optimal statistical rate in competing with the best covered policy:** In Theorem 1, we prove that our A-Crab algorithm, which uses average Bellman error, enjoys an optimal statistical rate of $1/\sqrt{N}$. In contrast, we prove in Proposition 4.1 that the ATAC algorithm, which uses the squared Bellman error, fails to achieve the optimal rate in certain offline learning instances. As Cheng et al. (2022) explains, the squared Bellman-error regularizer appears to be the culprit behind the suboptimal rate of ATAC being $1/N^{1/3}$. Moreover, our algorithm improves over any policy covered in the data. This is in contrast to the recent work Rashidinejad et al. (2022), which proposes an algorithm based on the MIS framework that achieves the $1/\sqrt{N}$ rate only when the optimal policy (i.e. the policy with the highest expected rewards) is covered in the data.

**2. A weaker notion of data coverage that exploits visitation structure:** As we discussed earlier, $\ell_\infty$ concentrability notion is used in many prior works (Rashidinejad et al., 2021; Chen and Jiang, 2022; Ozdaglar et al., 2022; Rashidinejad et al., 2022). Our importance-weighted average Bellman error as well as using Bernstein inequality in the proof relies on guarantees in terms of an $\ell_2$ single-policy concentrability notion that is weaker than the $\ell_\infty$ variant. In particular, we have $C_{\ell_\infty}^\pi = \|w^\pi\|_\infty$ and $C_{\ell_2}^\pi = \|w^\pi\|_{2,\mu}$, where $\|\cdot\|_{2,\mu}$ is the weighted 2-norm w.r.t. the dataset distribution $\mu$. The latter implies that the coverage coefficient only matters as much as it is covered by the dataset. Moreover, by the definition of $w^\pi$, we can obtain that $(C_{\ell_2}^\pi)^2 = \mathbb{E}_{d^\pi}[w^\pi(s,a)]$, which provides another explanation of $\ell_2$ concentrability that the coverage coefficient only matters as much as it is *actually* visited by the policy. There are also other notions of single-policy concentrability exploiting function approximation structures to make the coverage coefficient smaller (e.g., $C_{\text{Bellman}}^\pi$ in Xie et al. (2021)). However, these notions degenerate to $C_{\ell_\infty}^\pi$ as the function class gets richer (see Figure 1 for an intuitive comparison of different notions of concentrability and Section 2.4 for a rigorous proof).

**3. Robust policy improvement:** Policy improvement (PI) refers to the property that an offline RL algorithm (under a careful choice of specific hyperparameters) can always improve upon the behavior policy that is used to collect the data. In particular, robust policy improvement (RPI) means the PI property holds under a wide range of the choice of specific hyperparameters (Cheng et al., 2022; Xie et al., 2022). Similar to the ATAC algorithm in Cheng et al. (2022), our approach enjoys the robust policy improvement guarantee as shown in Theorem 2.

**4. Inheriting many other benefits of adversarially-trained actor-critic:** Since our algorithm is based on ATAC with a different choice of regularizer, it can be easily implemented as ATAC to be applied to practical scenarios, and we provide experimental results in Section 6. Also, our algorithm is robust to model misspecification and does not require the completeness assumption (Assumption 2 in Cheng et al. (2022)) on the value function class, which makes it more practical. Moreover, our algorithm can learn a policy that outperforms any other policies well covered by the dataset.

## 2   Background

**Notation.** We use $\Delta(\mathcal{X})$ to denote the probability simplex over a set $\mathcal{X}$ and use $\mathsf{Unif}(\mathcal{X})$ to denote the uniform distribution over $\mathcal{X}$. We denote by $\|\cdot\|_{2,\mu} = \sqrt{\mathbb{E}_\mu[(\cdot)^2]}$ the Euclidean norm weighted

by distribution $\mu$. We use the notation $x \lesssim y$ when there exists constant $c > 0$ such that $x \leq cy$ and $x \gtrsim y$ if $y \lesssim x$ and denote $x \asymp y$ if $x \lesssim y$ and $y \lesssim x$ hold simultaneously. We also use standard $O(\cdot)$ notation to hide constants and use $\tilde{O}(\cdot)$ to suppress logarithmic factors.

## 2.1 Markov decision process

An infinite-horizon discounted MDP is described by a tuple $M = (\mathcal{S}, \mathcal{A}, P, R, \gamma, \rho)$, where $\mathcal{S}$ is the state space, $\mathcal{A}$ is the action space, $P : \mathcal{S} \times \mathcal{A} \to \Delta(\mathcal{S})$ is the transition kernel, $R : \mathcal{S} \times \mathcal{A} \to \Delta([0,1])$ encodes a family of reward distributions with $r : \mathcal{S} \times \mathcal{A} \to [0,1]$ as the expected reward function, $\gamma \in [0,1)$ is the discount factor and $\rho : \mathcal{S} \to [0,1]$ is the initial state distribution. We assume $\mathcal{A}$ is finite while allowing $\mathcal{S}$ to be arbitrarily complex. A stationary (stochastic) policy $\pi : \mathcal{S} \to \Delta(\mathcal{A})$ specifies a distribution over actions in each state. Each policy $\pi$ induces a (discounted) occupancy density over state-action pairs $d^\pi : \mathcal{S} \times \mathcal{A} \to [0,1]$ defined as $d^\pi(s,a) := (1-\gamma)\sum_{t=0}^{\infty} \gamma^t P_t(s_t = s, a_t = a; \pi)$, where $P_t(s_t = s, a_t = a; \pi)$ denotes the visitation probability of state-action pair $(s,a)$ at time step $t$, starting at $s_0 \sim \rho(\cdot)$ and following $\pi$. We also write $d^\pi(s) = \sum_{a \in \mathcal{A}} d^\pi(s,a)$ to denote the (discounted) state occupancy, and use $\mathbb{E}_\pi[\cdot]$ as a shorthand of $\mathbb{E}_{(s,a) \sim d^\pi}[\cdot]$ or $\mathbb{E}_{s \sim d^\pi}[\cdot]$.

The value function of a policy $\pi$ is the discounted cumulative reward gained by executing that policy $V^\pi(s) := \mathbb{E}\left[\sum_{t=0}^{\infty} \gamma^t r_t \mid s_0 = s, a_t \sim \pi(\cdot|s_t), \forall t \geq 0\right]$ starting at state $s \in \mathcal{S}$ where $r_t = R(s_t, a_t)$. Similarly, we define $Q$ function of a policy as the expected cumulative reward gained by executing that policy starting from a state-action pair $(s,a)$, i.e., $Q^\pi(s,a) := \mathbb{E}\left[\sum_{t=0}^{\infty} \gamma^t r_t \mid s_0 = s, a_0 = a, a_t \sim \pi(\cdot|s_t), \forall t > 0\right]$. We write $J(\pi) := (1-\gamma)\mathbb{E}_{s \sim \rho}[V^\pi(s)] = \mathbb{E}_{(s,a) \sim d^\pi}[r(s,a)]$ to represent the (normalized) average value of policy $\pi$. We denote by $\pi^\star$ the optimal policy that maximizes the above objective and use the shorthand $V^\star := V^{\pi^\star}, Q^\star := Q^{\pi^\star}$ to denote the optimal value function and optimal $Q$ function respectively.

## 2.2 Function approximation

In modern RL, the state space $\mathcal{S}$ is usually large or infinite, making the classic tabular RL algorithms not scalable since their sample complexity depends on the cardinality of $\mathcal{S}$. Therefore, (general) function approximation is necessary for real-world scenarios with huge state space. In this paper, we assume access to three function classes: a function class $\mathcal{F} \subseteq \{f : \mathcal{S} \times \mathcal{A} \to [0, V_{\max}]\}$ that models the (approximate) Q-functions of policies, a function class $\mathcal{W} \subseteq \{w : \mathcal{S} \times \mathcal{A} \to [0, B_w]\}$[1] that represents marginalized importance weights with respect to data distribution, and a policy class $\Pi \subseteq \{\pi : \mathcal{S} \to \Delta(\mathcal{A})\}$ consisting of candidate policies. Our framework combines the marginalized importance sampling framework (e.g., Zhan et al. (2022); Rashidinejad et al. (2022)) with *actor-critic* methods (Xie et al., 2021; Cheng et al., 2022), which improves and selects among a set of candidate policies by successive computation of their Q-functions.

For any function $f \in \mathcal{F}$ and any policy $\pi \in \Pi$, we denote $f(s, \pi) = \sum_{a \in \mathcal{A}} \pi(a|s) f(s,a)$ for any $s \in \mathcal{S}$ and denote Bellman operator $\mathcal{T}^\pi : \mathbb{R}^{\mathcal{S} \times \mathcal{A}} \to \mathbb{R}^{\mathcal{S} \times \mathcal{A}}$ as

$$(\mathcal{T}^\pi f)(s,a) = r(s,a) + \gamma \mathbb{E}_{s' \sim P(\cdot|s,a)}[f(s', \pi)]. \tag{1}$$

Note that solving the fixed point equation (1) for $f$ finds the $Q$-function of policy $\pi$.

We make the following assumption on the expressivity of our function classes.

**Assumption 1** (Approximate Realizability). *Assume there exists $\epsilon_\mathcal{F} \geq 0$, s.t. for any policy $\pi \in \Pi$, $\min_{f \in \mathcal{F}} \max_{admissible\ \nu} \|f - \mathcal{T}^\pi f\|_{2,\nu}^2 \leq \epsilon_\mathcal{F}$, where admissible $\nu$ is defined by $\nu \in \{d^\pi | \pi \in \Pi\}$.*

This assumption is also required for Xie et al. (2021); Cheng et al. (2022). Note that when $\|f - \mathcal{T}^\pi f\|_{2,\nu}$ is small for all admissible $\nu$, we have $f \approx Q^\pi$. Therefore, Assumption 1 assumes that for any policy $\pi$, $Q^\pi$ is "approximatly" realized in $\mathcal{F}$. In particular, when $\epsilon_\mathcal{F} = 0$, Assumption 1 is equivalent to $Q^\pi \in \mathcal{F}$ for any $\pi \in \Pi$.

## 2.3 Offline reinforcement learning

In this paper, we study offline RL where we assume access only to a previously-collected and fixed dataset of interactions $\mathcal{D} = \{(s_i, a_i, r_i, s_i')\}_{i=1}^N$, where $r_i \sim R(s_i, a_i)$, $s_i' \sim P(\cdot \mid s_i, a_i)$.

---

[1]Without loss of generality, we always assume that the all-one function is contained in $\mathcal{W}$.

To streamline the analysis, we assume that $(s_i, a_i)$ pairs are generated i.i.d. according to a data distribution $\mu \in \Delta(\mathcal{S} \times \mathcal{A})$. We make the common assumption that the dataset is collected by a behavior policy, i.e., $\mu$ is the discounted visitation probability of a behavior policy, which we also denote by $\mu$. For convenience, we assume the behavior policy $\mu \in \Pi$. The goal of offline RL is to learn a *good* policy $\hat{\pi}$ (a policy with a high $J(\hat{\pi})$) using the offline dataset. Also, for any function $f$ that takes $(s, a, r, s')$ as input, we define the expectation w.r.t. the dataset $\mathcal{D}$ (or empirical expectation) as $\mathbb{E}_{\mathcal{D}}[f] = \frac{1}{N} \sum_{(s_i, a_i, r_i, s'_i) \in \mathcal{D}} f(s_i, a_i, r_i, s'_i)$.

**Marginalized importance weights.** We define the marginalized importance weights of any policy $\pi$ to be the ratio between the discounted state-action occupancy of $\pi$ and the data distribution $w^{\pi}(s, a) := \frac{d^{\pi}(s,a)}{\mu(s,a)}$. Such weights have been defined in prior works on theoretical RL (Xie and Jiang, 2020; Zhan et al., 2022; Rashidinejad et al., 2022; Ozdaglar et al., 2022) as well as practical RL algorithms (Nachum et al., 2019a,b; Zhang et al., 2020b,c; Lee et al., 2021).

## 2.4 Coverage of offline dataset

We study offline RL with access to a dataset with partial coverage. We measure the coverage of policy $\pi$ in the dataset using the weighted $\ell_2$ single-policy concentrability coefficient defined below.

**Definition 1** ($\ell_2$ concentrability). *Given a policy $\pi$, define $C_{\ell_2}^{\pi} = \|w^{\pi}\|_{2,\mu} = \|d^{\pi}/\mu\|_{2,\mu}$.*

This definition is much weaker than the all-policy concentrability conventionally used in offline RL (Scherrer, 2014; Liu et al., 2019a; Chen and Jiang, 2019; Jiang, 2019; Wang et al., 2019; Liao et al., 2020; Zhang et al., 2020a), which requires the ratio $\frac{d^{\pi}(s,a)}{\mu(s,a)}$ to be bounded for all $s \in \mathcal{S}$ and $a \in \mathcal{A}$ as well as all policies $\pi$. The following proposition compares two variants of single-policy concentrability definition that appeared in recent works Rashidinejad et al. (2021); Xie et al. (2021) with the $\ell_2$ variant defined in Definition 1; see Appendix A.1 for more discussion on different concentrability definitions in prior work. To our knowledge, the $\ell_2$ version of concentrability definition has been only used in offline RL with all-policy coverage (Farahmand et al., 2010; Xie and Jiang, 2020). In the context of partial coverage, Uehara and Sun (2021) used $\ell_2$ version in a model-based setting, but their algorithms are computationally intractable. Recent works Xie et al. (2021); Cheng et al. (2022) use another milder version of concentrability than $\ell_{\infty}$, and we compare different concentrability versions in Proposition 2.1. An intuitive comparison is presented in Figure 1.

**Proposition 2.1** (Comparing concentrability definitions). *Define the $\ell_{\infty}$ single-policy concentrability (Rashidinejad et al., 2021) as $C_{\ell_{\infty}}^{\pi} = \|d^{\pi}/\mu\|_{\infty}$ and the Bellman-consistent single-policy concentrability (Xie et al., 2021) as $C_{Bellman}^{\pi} = \max_{f \in \mathcal{F}} \frac{\|f - \mathcal{T}^{\pi} f\|_{2,d^{\pi}}^2}{\|f - \mathcal{T}^{\pi} f\|_{2,\mu}^2}$. Then, it always holds $(C_{\ell_2}^{\pi})^2 \leq C_{\ell_{\infty}}^{\pi}$, $C_{\ell_2}^{\pi} \leq C_{\ell_{\infty}}^{\pi}$ and there exist offline RL instances where $(C_{\ell_2}^{\pi})^2 \leq C_{Bellman}^{\pi}$, $C_{\ell_2}^{\pi} \leq C_{Bellman}^{\pi}$.*

A proof for Proposition 2.1 is presented in Appendix B. It is easy to show that the $\ell_2$ variant is bounded by $\ell_{\infty}$ variant of concentrability as the former requires $\mathbb{E}_{d^{\pi}}[w^{\pi}(s, a)]$ to be bounded while the latter requires $w^{\pi}(s, a)$ to be bounded for any $s \in \mathcal{S}$ and $a \in \mathcal{A}$. Example 1 provides a concrete example that $C_{\ell_2}^{\pi}$ is bounded by a constant while $C_{\ell_{\infty}}^{\pi}$ could be arbitrarily large.

**Example 1** (Arbitrarily large $\ell_{\infty}$ concentrability with a constant $\ell_2$ concentrability). *Consider the simplest two-arm bandit settings, where the dataset distribution is $\mu(a_1) = 1 - \epsilon^2$, $\mu(a_2) = \epsilon^2$ for an arbitrarily small $\epsilon > 0$. Let $\pi$ be a policy s.t. $\pi(a_1) = d^{\pi}(a_1) = 1 - \epsilon$, $\pi(a_2) = d^{\pi}(a_2) = \epsilon$. Then one can calculate that $w^{\pi}(a_1) = \frac{1-\epsilon}{1-\epsilon^2} \leq 1$ and $w^{\pi}(a_2) = \frac{1}{\epsilon}$. Therefore, $C_{\ell_2}^{\pi} \leq \sqrt{2}$ while $C_{\ell_{\infty}}^{\pi} = \frac{1}{\epsilon}$ can be arbitrarily large.*

Furthermore, the Bellman-consistent variant can exploit the structure in the Q-function class $\mathcal{F}$ for a smaller concentrability coefficient. However, in situations where the class $\mathcal{F}$ is highly expressive, $C_{Bellman}^{\pi}$ could be close to $C_{\ell_{\infty}}^{\pi}$ and thus possibly larger than $C_{\ell_2}^{\pi}$.

Finally, we make a boundedness assumption on our marginalized importance weight function class $\mathcal{W}$ in terms of $\ell_2$ concentrability and a single-policy realizability assumption.

**Assumption 2** (Boundedness in $\ell_2$ norm of $\mathcal{W}$). *Assume $\|w\|_{2,\mu} \leq C_{\ell_2}^{\star}$ for all $w \in \mathcal{W}$.*

**Assumption 3** (Single-policy realizability of $w^{\pi}$). *Assume $w^{\pi} \in \mathcal{W}$ for some policy $\pi \in \Pi$ that we aim to compete with.*

The definition of $C^\star_{\ell_2}$ is similar to Xie and Jiang (2020) but they need $w^\pi \in \mathcal{W}$ for all $\pi \in \Pi$, which is much stronger than our single-policy realizability assumption for $\mathcal{W}$.

# 3 Actor-Critic Regularized by Average Bellman Error

In this section, we introduce our main algorithm named A-Crab (**A**ctor-**C**ritic **R**egularized by **A**verage **B**ellman error, Algorithm 1), and compare it with the previous ATAC algorithm (Cheng et al., 2022). In Section 4, we will provide theoretical guarantees of A-Crab and discuss its advantages.

## 3.1 From Actor-Critic to A-Crab

Our algorithm design builds upon the actor-critic method, in which we iteratively evaluate a policy and improve the policy based on the evaluation. Consider the following actor-critic example:

$$\hat{\pi}^* \in \arg\max_{\pi \in \Pi} f^\pi(s_0, \pi), \quad f^\pi \in \arg\min_{f \in \mathcal{F}} \mathbb{E}_\mu[((f - \mathcal{T}^\pi f)(s, a))^2],$$

where we assume $s_0$ is the fixed initial state in this example and recall that $f(s, \pi) = \sum_{a \in \mathcal{A}} \pi(a|s) f(s, a)$. Here, the policy is evaluated by the function that minimizes the squared Bellman error. However, insufficient data coverage may lead the critic to give an unreliable evaluation of the policy. To address this, the critic can compute a *Bellman-consistent* pessimistic evaluation of $\pi$ (Xie et al., 2021), which picks the most pessimistic $f \in \mathcal{F}$ that approximately satisfies the Bellman equation. Introducing a hyperparameter $\beta \geq 0$ to tradeoff between pessimism and Bellman consistency yields the following criteria for the critic:

$$f^\pi \in \arg\min_{f \in \mathcal{F}} f(s_0, \pi) + \beta \mathbb{E}_\mu[((f - \mathcal{T}^\pi f)(s, a))^2].$$

Cheng et al. (2022) argue that instead of the above *absolute pessimism*, a *relative pessimism* approach of optimizing the performance of $\pi$ *relative* to the behavior policy, results in an algorithm that improves over the behavior policy for any $\beta \geq 0$ (i.e., robust policy improvement). Incorporating relative pessimism in the update rule gives the ATAC algorithm (Cheng et al., 2022):

$$\hat{\pi}^* \in \arg\max_{\pi \in \Pi} \mathbb{E}_\mu[f^\pi(s, \pi) - f^\pi(s, a)],$$
$$f^\pi \in \arg\min_{f \in \mathcal{F}} \mathbb{E}_\mu[f(s, \pi) - f(s, a)] + \beta \mathbb{E}_\mu[((f - \mathcal{T}^\pi f)(s, a))^2].$$

Finally, we introduce the importance weights $w(s, a)$ and change the squared Bellman regularizer to an importance-weighted average Bellman error to arrive at:

$$\hat{\pi}^\star \in \arg\max_{\pi \in \Pi} \mathcal{L}_\mu(\pi, f^\pi), \quad \text{s.t. } f^\pi \in \arg\min_{f \in \mathcal{F}} \mathcal{L}_\mu(\pi, f) + \beta \mathcal{E}_\mu(\pi, f), \tag{2}$$

where

$$\mathcal{L}_\mu(\pi, f) = \mathbb{E}_\mu[f(s, \pi) - f(s, a)], \tag{3}$$
$$\mathcal{E}_\mu(\pi, f) = \max_{w \in \mathcal{W}} |\mathbb{E}_\mu[w(s, a)(f - \mathcal{T}^\pi f)(s, a)]|. \tag{4}$$

Maximization over $w$ in the importance-weighted average Bellman regularizer in (4) ensures that the Bellman error is small when averaged over measure $\mu \cdot w$ for any $w \in \mathcal{W}$, which turns out to be sufficient to control the suboptimality of the learned policy as the performance difference decomposition Lemma 1 shows. [2]

**Squared Bellman error v.s. importance-weighted average Bellman error.** Unlike our approach, the ATAC algorithm in Cheng et al. (2022) uses squared Bellman error, wherein direct empirical approximation leads to overestimating the regularization term.[3] To obtain an unbiased empirical estimator, Cheng et al. (2022) uses $\mathbb{E}_\mathcal{D}\left[(f(s, a) - r - \gamma f(s', \pi))^2\right] - \min_{g \in \mathcal{F}} \mathbb{E}_\mathcal{D}\left[(g(s, a) - r - \gamma f(s', \pi))^2\right]$ as the empirical estimator which subtracts the overestimation. Yet, as we later see in Proposition 4.1, even with this correction, ATAC fails to achieve the optimal statistical rate of $1/\sqrt{N}$ in certain offline learning instances. In contrast, the importance-weighted average Bellman error in our algorithm is unbiased (as it involves no non-linearity). This makes our theoretical analysis much simpler and leads to achieving an optimal statistical rate of $1/\sqrt{N}$ as shown in Theorem 1.

---

[2] Such importance-weighted minimax formulations of Bellman error have been used in prior work on off-policy evaluation (Uehara et al., 2020) and offline RL with all-policy coverage (Xie and Jiang, 2020).

[3] This is closely related to the infamous double-sampling issue; see Section 3.1 in Chen and Jiang (2019) for a detailed discussion.

## 3.2 Main algorithms

Since we do not have direct access to the dataset distribution $\mu$, our algorithm instead solves an empirical version of (2), which can be formalized as

$$\hat{\pi} \in \arg\max_{\pi \in \Pi} \mathcal{L}_{\mathcal{D}}(\pi, f^{\pi}), \quad \text{s.t.} \ f^{\pi} \in \arg\min_{f \in \mathcal{F}} \mathcal{L}_{\mathcal{D}}(\pi, f) + \beta \mathcal{E}_{\mathcal{D}}(\pi, f), \tag{5}$$

where

$$\mathcal{L}_{\mathcal{D}}(\pi, f) = \mathbb{E}_{\mathcal{D}}[f(s, \pi) - f(s, a)], \tag{6}$$

$$\mathcal{E}_{\mathcal{D}}(\pi, f) = \max_{w \in \mathcal{W}} |\mathbb{E}_{\mathcal{D}}[w(s, a)(f(s, a) - r - \gamma f(s', \pi))]| . \tag{7}$$

---

**Algorithm 1** **A**ctor-**C**ritic **R**egularized by **A**verage **B**ellman error (A-CRAB)

---

1: **Input:** Dataset $\mathcal{D} = \{(s_i, a_i, r_i, s_i')\}_{i=1}^{N}$, value function class $\mathcal{F}$, importance weight function class $\mathcal{W}$, no-regret policy optimization oracle PO (Definition 2).
2: Initialization: $\pi_1$ : uniform policy, $\beta$: hyperparameter.
3: **for** $k = 1, 2, \ldots, K$ **do**
4:     $f_k \leftarrow \arg\min_{f \in \mathcal{F}} \mathcal{L}_{\mathcal{D}}(\pi_k, f) + \beta \mathcal{E}_{\mathcal{D}}(\pi_k, f)$, where $\mathcal{L}_{\mathcal{D}}$ and $\mathcal{E}_{\mathcal{D}}$ are defined in (6), (7)
5:     $\pi_{k+1} \leftarrow \text{PO}(\pi_k, f_k, \mathcal{D})$.
6: **end for**
7: **Output:** $\bar{\pi} = \text{Unif}\left(\{\pi_k\}_{k=1}^{K}\right)$.

---

Similar to Cheng et al. (2022), we view program (5) as a Stackelberg game and solve it using a no-regret oracle as shown in Algorithm 1. At each step $k$, the critic minimizes the objective defined by (7) w.r.t. $\pi_k$, and $\pi_{k+1}$ is generated by a no-regret policy optimization oracle, given below.

**Definition 2** (No-regret policy optimization oracle)**.** *An algorithm PO is defined as a no-regret policy optimization oracle if for any (adversarial) sequence of functions $f_1, f_2, \ldots, f_K \in \mathcal{F}$ where $f_k : \mathcal{S} \times \mathcal{A} \to [0, V_{\max}], \forall k \in [K]$, the policy sequence $\pi_1, \pi_2, \ldots, \pi_K$ produced by PO satisfies that for any policy $\pi \in \Pi$, it holds that $\epsilon_{opt}^{\pi} \triangleq \frac{1}{K} \sum_{k=1}^{K} \mathbb{E}_{\pi}[f_k(s, \pi) - f_k(s, \pi_k)] = o(1)$.*

Among the well-known instances of the above no-regret policy optimization oracle is natural policy gradient (Kakade, 2001) of the form $\pi_{k+1}(a|s) \propto \pi_k(a|s) \exp(\eta f_k(s, a))$ with $\eta = \sqrt{\frac{\log |\mathcal{A}|}{2V_{\max}^2 K}}$ (Even-Dar et al., 2009; Agarwal et al., 2021; Xie et al., 2021; Cheng et al., 2022). A detailed discussion of the above policy optimization oracle can be found in Cheng et al. (2022). Utilizing the no-regret oracle in solving the Stackelberg optimization problem in (5) yields Algorithm 1.

**A remark on critic's optimization problem.** In our algorithm, for any given $\pi$, the critic needs to solve a $\min_{f \in \mathcal{F}} \max_{w \in \mathcal{W}}$ optimization problem, whereas in ATAC (Cheng et al., 2022), the critic needs to solve a $\min_{f \in \mathcal{F}} \max_{g \in \mathcal{F}}$ problem. Since we only assume single-policy realizability for the class $\mathcal{W}$ (Assumption 3) but assume all-policy realizability for $\mathcal{F}$ (Assumption 1) (and Cheng et al. (2022) even requires the Bellman-completeness assumption over $\mathcal{F}$ which is much stronger), in general, the cardinality of $\mathcal{W}$ could be much smaller than $\mathcal{F}$, which makes the optimization region of the critic's optimization problem in our algorithm $\mathcal{F} \times \mathcal{W}$ smaller than $\mathcal{F} \times \mathcal{F}$ in ATAC.

## 4 Theoretical Analysis

In this section, we show the theoretical guarantee of our main algorithm (Algorithm 1), which is statistically optimal in terms of $N$.

### 4.1 Performance guarantee of the A-Crab algorithm

We first formally present our main theorem, which provides a theoretical guarantee of our A-Crab algorithm (Algorithm 1). A proof sketch is provided in Section 5 and the complete proof is deferred to Appendix C.3.

**Theorem 1** (Main theorem). *Under Assumptions 1 and 2 and let $\pi \in \Pi$ be any policy satisfying Assumption 3, then with probability at least $1 - \delta$,*

$$J(\pi) - J(\bar{\pi}) \leq O\left(\epsilon_{stat} + C^{\star}_{\ell_2}\sqrt{\epsilon_{\mathcal{F}}}\right) + \epsilon^{\pi}_{opt},$$

*where $\epsilon_{stat} \asymp V_{\max}C^{\star}_{\ell_2}\sqrt{\frac{\log(|\mathcal{F}||\Pi||\mathcal{W}|/\delta)}{N}} + \frac{V_{\max}B_w \log(|\mathcal{F}||\Pi||\mathcal{W}|/\delta)}{N}$, and $\bar{\pi}$ is returned by Algorithm 1 with the choice of $\beta = 2$.*

Below we discuss the advantages of our approach as shown in the above theorem.

**Optimal statistical rate and computational efficiency.** When $\epsilon_{\mathcal{F}} = 0$ (i.e., there is no model misspecification), and when $\pi = \pi^{\star}$ is one of the optimal policies, the output policy $\bar{\pi}$ achieves $O(1/\sqrt{N})$ suboptimality rate which is optimal in $N$ dependence (as long as $K$ is large enough). This improves the $O(1/N^{1/3})$ rate of the previous algorithm (Cheng et al., 2022). Note that the algorithm of Xie et al. (2021) can also achieve the optimal $O(1/\sqrt{N})$ rate but their algorithm involves hard constraints of squared $\ell_2$ Bellman error and thus is computationally intractable. Cheng et al. (2022) convert the hard constraints to a regularizer, making the algorithm computationally tractable while degenerating the statistical rate. Our algorithm is both statistically optimal and computationally efficient, which improves upon both Xie et al. (2021); Cheng et al. (2022) simultaneously.

**Competing with any policy.** Another advantage of our algorithm is that it can compete with any policy $\pi \in \Pi$ as long as $w^{\pi} = d^{\pi}/\mu$ is contained in $\mathcal{W}$. In particular, the importance ratio of the behavior policy $w^{\mu} = d^{\mu}/\mu = \mu/\mu \equiv 1$ is always contained in $\mathcal{W}$, which implies that our algorithm satisfies robust policy improvement (see Theorem 2 for details).

**Robustness to model misspecification.** Theorem 1 also shows that our algorithm is robust to model misspecification on realizability assumption. Note that our algorithm does not need a completeness assumption, while Xie et al. (2021); Cheng et al. (2022) both require the (approximate) completeness assumption.

**Removal of the completeness assumption on $\mathcal{F}$.** Compared to our algorithm, Cheng et al. (2022) additionally need a completeness assumption on $\mathcal{F}$, which requires that for any $f \in \mathcal{F}$ and $\pi \in \Pi$, it approximately holds that $\mathcal{T}^{\pi}f \in \mathcal{F}$. They need this completeness assumption because they use the estimator $\mathbb{E}_{\mathcal{D}}\left[(f(s,a) - r - \gamma f(s',\pi))^2\right] - \min_{g \in \mathcal{F}} \mathbb{E}_{\mathcal{D}}\left[(g(s,a) - r - \gamma f(s',\pi))^2\right]$ to address the over-estimation issue caused by their squared $\ell_2$ Bellman error regularizer, and to make this estimator accurate, they need $\min_{g \in \mathcal{F}} \mathbb{E}_{\mathcal{D}}\left[(g(s,a) - r - \gamma f(s',\pi))^2\right]$ to be small, which can be implied by the (approximate) completeness assumption. In our algorithm, thanks to the nice property of the weighted average Bellman error regularizer which can be estimated by a simple and unbiased estimator, we can get rid of this strong assumption.

### 4.2 A-Crab for robust policy improvement

Robust policy improvement (RPI) refers to the property of an offline RL algorithm that the learned policy (almost) always improves upon the behavior policy used to collect data over a wide range of the choice of some specific hyperparameters (in this paper, the hyperparameter is $\beta$) (Cheng et al., 2022). Similar to ATAC in Cheng et al. (2022), our A-Crab also enjoys the RPI property. Theorem 2 implies that as long as $\beta = o(\sqrt{N})$, our algorithm can learn a policy with vanishing suboptimality compared to the behavior policy with high probability. The proof is deferred to Appendix D.

**Theorem 2** (Robust policy improvement). *Under Assumptions 1 and 2, with probability at least $1 - \delta$,*

$$J(\mu) - J(\bar{\pi}) \lesssim (\beta + 1)(\epsilon_{stat} + C^{\star}_{\ell_2}\sqrt{\epsilon_{\mathcal{F}}}) + \epsilon^{\pi}_{opt},$$

*where $\epsilon_{stat} \asymp V_{\max}C^{\star}_{\ell_2}\sqrt{\frac{\log(|\mathcal{F}||\Pi||\mathcal{W}|/\delta)}{N}} + \frac{V_{\max}B_w \log(|\mathcal{F}||\Pi||\mathcal{W}|/\delta)}{N}$, and $\bar{\pi}$ is returned by Algorithm 1 with the choice of any $\beta \geq 0$.*

### 4.3 Suboptimality of squared $l_2$ norm of Bellman error as regularizers

The ATAC algorithm of Cheng et al. (2022) suffers suboptimal statistical rate $O(1/N^{1/3})$ due to the squared $\ell_2$ Bellman error regularizer. Intuitively, in Cheng et al. (2022), they use Lemma 1 to

decompose the performance difference and use $\|f - \mathcal{T}^\pi f\|_{2,\mu}$ to upper bound $\mathbb{E}_\mu[(f - \mathcal{T}^\pi f)(s,a)]$, which causes suboptimality since in general the former could be much larger than the latter. To overcome this suboptimal step, in our algorithm, we use a weighted version of $\mathbb{E}_\mu[(f - \mathcal{T}^\pi f)(s,a)]$ as our regularizer instead of $\|f - \mathcal{T}^\pi f\|_{2,\mu}$. Proposition 4.1 shows that ATAC is indeed statistically suboptimal even under their optimal choice of the hyperparameter $\beta = \Theta(N^{2/3})$. The proof is deferred to Appendix E.1.

**Proposition 4.1** (Suboptimality of ATAC). *If we change the regularizer s.t.*

$$\mathcal{E}_\mu(\pi, f) = \|f - \mathcal{T}^\pi f\|_{2,\mu}^2 \quad \text{and} \quad \mathcal{E}_\mathcal{D}(\pi, f) = \mathcal{L}(f, f, \pi, \mathcal{D}) - \min_{g \in \mathcal{F}} \mathcal{L}(g, f, \pi, \mathcal{D}),$$

*where $\mathcal{L}(g, f, \pi, \mathcal{D}) = \mathbb{E}_\mathcal{D}[(g(s,a) - r - \gamma f(s', \pi))^2]$, then even under all policy realizability ($Q^\pi \in \mathcal{F}$ for all $\pi \in \Pi$) and completeness assumption ($\mathcal{T}^\pi f \in \mathcal{F}$ for all $\pi \in \Pi$ and $f \in \mathcal{F}$), with their optimal choice of $\beta = \Theta(N^{2/3})$ (Cheng et al., 2022), there exists an instance s.t. the suboptimality of the returned policy of (5) (i.e., the output policy by ATAC) is $\Omega(1/N^{1/3})$ with at least constant probability.*

# 5 Proof Sketch

We provide a proof sketch of our main theorem (Theorem 1) in this section. The key lemma of the proof is presented in Lemma 1.

**Lemma 1** (Performance difference decomposition, Lemma 12 in Cheng et al. (2022)). *For any $\pi, \hat{\pi} \in \Pi$, and any $f : \mathcal{S} \times \mathcal{A} \to \mathbb{R}$, we can decompose $J(\pi) - J(\hat{\pi})$ as*

$$\mathbb{E}_\mu[(f - \mathcal{T}^{\hat{\pi}} f)(s,a)] + \mathbb{E}_\pi[(\mathcal{T}^{\hat{\pi}} f - f)(s,a)] + \mathbb{E}_\pi[f(s,\pi) - f(s,\hat{\pi})] + \mathcal{L}_\mu(\hat{\pi}, f) - \mathcal{L}_\mu(\hat{\pi}, Q^{\hat{\pi}}).$$

Note that the first two terms in the RHS of the decomposition are average Bellman errors of $f$ and $\hat{\pi}$ w.r.t. $\mu$ and $d^\pi$. Based on this lemma, we can directly use the average Bellman error as the regularizer instead of the squared Bellman error, which could be much larger and causes suboptimality.

*Proof sketch of Theorem 1.* For simplicity, we assume realizability of $Q^\pi$, i.e., $\epsilon_\mathcal{F} = 0$. By Lemma 1 and the definition of $\bar{\pi}$, we have $J(\pi) - J(\bar{\pi}) = \frac{1}{K} \sum_{k=1}^K (J(\pi) - J(\pi_k))$, which equals to

$$\frac{1}{K} \sum_{k=1}^K (\underbrace{\mathbb{E}_\mu[(f_k - \mathcal{T}^{\pi_k} f_k)(s,a)]}_{(a)} + \underbrace{\mathbb{E}_\pi[(\mathcal{T}^{\pi_k} f_k - f_k)(s,a)]}_{(b)}$$
$$+ \underbrace{\mathbb{E}_\pi[f_k(s,\pi) - f_k(s,\pi_k)]}_{(c)} + \underbrace{\mathcal{L}_\mu(\pi_k, f_k) - \mathcal{L}_\mu(\pi_k, Q^{\pi_k})}_{(d)}).$$

By the concentration argument, with high probability, we have $\mathcal{E}_\mu(\pi, f) = \mathcal{E}_\mathcal{D}(\pi, f) \pm \epsilon_{\text{stat}}$ and $\mathcal{L}_\mu(\pi, f) = \mathcal{L}_\mathcal{D}(\pi, f) \pm \epsilon_{\text{stat}}$ for all $\pi \in \Pi$ and $f \in \mathcal{F}$. Combining the fact that $d^\pi/\mu \in \mathcal{W}$, one can show that $(a) + (b) \leq 2\mathcal{E}_\mu(\pi_k, f_k) \leq 2\mathcal{E}_\mathcal{D}(\pi_k, f_k) + O(\epsilon_{\text{stat}})$. Therefore,

$$(a) + (b) + (d) \leq \mathcal{L}_\mu(\pi_k, f_k) + 2\mathcal{E}_\mathcal{D}(\pi_k, f_k) + O(\epsilon_{\text{stat}}) - \mathcal{L}_\mu(\pi_k, Q^{\pi_k})$$
$$\leq \mathcal{L}_\mathcal{D}(\pi_k, f_k) + 2\mathcal{E}_\mathcal{D}(\pi_k, f_k) + O(\epsilon_{\text{stat}}) - \mathcal{L}_\mathcal{D}(\pi_k, Q^{\pi_k})$$
$$\leq \mathcal{L}_\mathcal{D}(\pi_k, Q^{\pi_k}) + 2\mathcal{E}_\mathcal{D}(\pi_k, Q^{\pi_k}) + O(\epsilon_{\text{stat}}) - \mathcal{L}_\mathcal{D}(\pi_k, Q^{\pi_k})$$
$$\leq O(\epsilon_{\text{stat}}) + 2\mathcal{E}_\mu(\pi_k, Q^{\pi_k}) = O(\epsilon_{\text{stat}}),$$

where the third inequality holds by the optimality of $f_k$, and the last equality holds since the Bellman error of $Q^\pi$ w.r.t. $\pi$ is 0. Therefore, with high probability,

$$J(\pi) - J(\bar{\pi}) \leq O(\epsilon_{\text{stat}}) + \epsilon_{\text{opt}}^\pi.$$

$\square$

# 6 Experiments

In this section, we conduct experiments of our proposed A-Crab algorithm (Algorithm 1) using a selection of the Mujoco datasets (v2) from D4RL offline RL benchmark (Fu et al., 2020). In particular, we compare the performances of A-Crab and ATAC, since ATAC is the state-of-the-art algorithm on a range of continuous control tasks (Cheng et al., 2022).

**A more practical version of weighted average Bellman error.** Recall the definition of our proposed weighted average Bellman error regularizer

$$\mathcal{E}_{\mathcal{D}}(\pi, f) = \max_{w \in \mathcal{W}} |\mathbb{E}_{\mathcal{D}}[w(s,a)(f(s,a) - r - \gamma f(s',\pi))]| .$$

Since the calculation of $\mathcal{E}_{\mathcal{D}}(\pi, f)$ requires solving an optimization problem w.r.t. importance weights $w$, for computational efficiency, we choose $\mathcal{W} = [0, C_\infty]^{\mathcal{S} \times \mathcal{A}}$ as in Hong et al. (2023), and thus

$$\mathcal{E}_{\mathcal{D}}^{\mathrm{app}}(\pi, f) = C_\infty \max\{\mathbb{E}_{\mathcal{D}}[(f(s,a) - r - \gamma f(s',\pi))_+], \mathbb{E}_{\mathcal{D}}[(r + \gamma f(s',\pi) - f(s,a))_+]\}, \quad (8)$$

where $(\cdot)_+ = \max\{\cdot, 0\}$ and $C_\infty$ can be viewed as a hyperparameter. We also observed that using a combination of squared Bellman error and our average Bellman error achieves better performance in practice, and we conjecture the reason is that the squared Bellman error regularizer is computationally more efficient and statistically suboptimal, while our average Bellman error regularizer is statistically optimal while computationally less efficient, and thus the combination of these two regularizers can benefit the training procedure.

The practical implementation of our algorithm is nearly identical to (a lightweight version of) ATAC (Cheng et al., 2022) , except that we choose

$$\frac{1}{2} \left( 2\mathcal{E}_{\mathcal{D}}^{\mathrm{app}}(\pi, f) + \beta \mathbb{E}_{\mathcal{D}}[((f - \mathcal{T}^\pi f)(s,a))^2] \right)$$

as the regularizer, while ATAC uses $\beta \mathbb{E}_{\mathcal{D}}[((f - \mathcal{T}^\pi f)(s,a))^2]$. All hyperparameters are the same as ATAC, including $\beta$. For the additional hyperparamter $C_\infty$, we do a grid search on $\{1, 2, 5, 10, 20, 50, 100, 200\}$.

Figure 2 compares the performance of ACrab and ATAC during training. It shows that our A-Crab has higher returns and smaller deviations than ATAC in various settings (*walker2d-random, halfcheetah-medium-replay, hopper-medium-expert*). We provide more details and results in Appendix F. We also observed that in most settings, A-Crab has a smaller variance, which shows that the training procedure of A-Crab is more stable. We provide the choice of $\beta$ and $C_\infty$ for each setting in Table F.3. Note that we use the same value of $\beta$ as in ATAC.

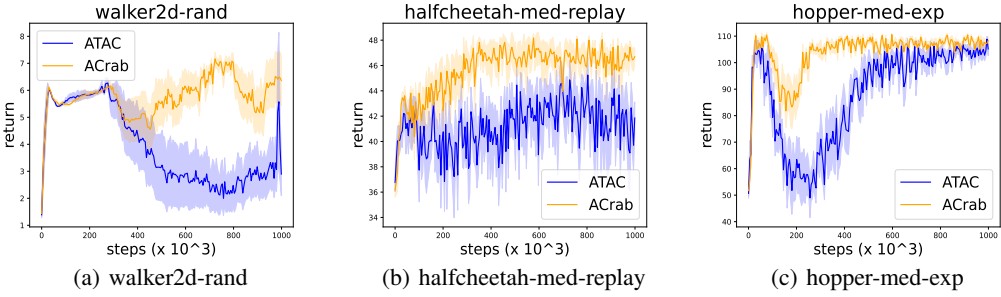

(a) walker2d-rand       (b) halfcheetah-med-replay       (c) hopper-med-exp

Figure 2: Comparison of A-Crab and ATAC. For each algorithm, we run 8 copies with random seeds 0-7 and plot the mean and standard deviation. We use the same pre-training method as ATAC for 100 epochs, and the plot starts after pre-training.

## 7 Discussion

We present a new offline RL algorithm called A-Crab (Algorithm 1) that can be combined with general function approximators and handle datasets with partial coverage. A-Crab is an actor-critic method, where the critic finds a relatively pessimistic evaluation of the actor while minimizing an importance-weighted average Bellman error. We prove that A-Crab achieves the optimal statistical rate of $1/\sqrt{N}$ converging to the best policy "covered" in the data. Importantly, the notion of *coverage* here is a weaker $\ell_2$ variant of the single-policy concentrability, which only requires the average marginalized importance weights over visitations of the target policy to be bounded. Also, A-Crab enjoys robust policy improvement that consistently improves over the data-collection behavior policy. Moreover, we empirically validated the effectiveness of A-Crab in the D4RL benchmark. Interesting avenues for future work include combining A-Crab's offline learning with an online fine-tuning algorithm with a limited trial-and-error budget and developing new measures for single-policy coverage that leverage both the visitation and hypothesis class structures.

## Acknowledgements

This work is partially supported by NSF Grants IIS-1901252 and CCF-1909499. The work was done when HZ was a visiting researcher at Meta.

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

# A Related Work

In this section, we review additional related literature not covered in the introductions.

## A.1 Dataset Coverage Assumptions

One central challenge of offline RL is the insufficient coverage of the dataset. In RL theory, concentrability is often used to characterize dataset coverage (Munos, 2007; Scherrer, 2014). For example, many earlier works require all-policy $\ell_\infty$ concentrability (Scherrer, 2014; Liu et al., 2019a; Chen and Jiang, 2019; Jiang, 2019; Wang et al., 2019; Liao et al., 2020; Zhang et al., 2020a). Some works even require the ratio between occupancy probability induced by polices and the dataset distribution to be bounded for every time step (Szepesvári and Munos, 2005; Munos, 2007; Antos et al., 2008; Farahmand et al., 2010; Antos et al., 2007). The work Xie and Jiang (2021) makes stronger assumptions such as requiring lower bound on conditionals $\mu(a|s)$.

Since the all-policy concentrability assumption is strong and can often be violated in practice, recent algorithms requiring only partial data coverage assumptions are developed based on single-policy $\ell_\infty$ concentrability (Rashidinejad et al., 2021; Zhan et al., 2022; Rashidinejad et al., 2022; Chen and Jiang, 2022; Ozdaglar et al., 2022). However, This could still be restrictive even if only single-policy concentrability is required since the $\ell_\infty$ concentrability is an upper bound of density ratios over all state-action pairs.

Milder versions of $\ell_\infty$ concentrability have been studied in both all-policy concentrability framework (Xie and Jiang, 2020, 2021; Feng et al., 2019; Uehara et al., 2020) or and single-policy concentrability (Uehara and Sun, 2021; Xie et al., 2021; Song et al., 2022; Cheng et al., 2022). However, these works based on milder versions of $\ell_\infty$ single-policy concentrability are either computationally intractable (Uehara and Sun, 2021; Xie et al., 2021) or suffer a suboptimal statistical rate (Cheng et al., 2022).

Our work uses $\ell_2$ concentrability version, which also appears in Xie and Jiang (2020). In particular, they also use weighted average Bellman error in their algorithm. However, their algorithm requires all-policy concentrability assumptions and thus cannot deal with partial dataset coverage. To the best of our knowledge among previous works, only Uehara and Sun (2021) used $\ell_2$ single-policy concentrability to characterize data coverage, but their algorithm is designed for model-based settings and are computationally intractable. Another closely related work is Uehara et al. (2020), which also uses weighted average Bellman error. However, their algorithm is in the off-policy evaluation (OPE) framework, and they use $\ell_\infty$ concentrability version to characterize dataset coverage.

## A.2 Conservative offline reinforcement learning

To address partial dataset coverage in offline RL, a line of recent applied works studies conservative algorithms, which can be divided into several categories.

The first category enforces the learned policy to be close to the behavior policy (or equivalently, dataset), which ensures that candidate policies not well covered by the dataset are eliminated. This can be accomplished by either adding constraints explicitly (Fujimoto et al., 2019; Kumar et al., 2019; Wu et al., 2019; Jaques et al., 2019; Siegel et al., 2020; Ghasemipour et al., 2020; Fujimoto and Gu, 2021), implicitly (Peng et al., 2019; Nair et al., 2020), or by importance sampling with bounded ratio (Swaminathan and Joachims, 2015; Liu et al., 2019b; Nachum et al., 2019b; Zhang et al., 2020c,b; Lee et al., 2021).

The second category consists of model-based methods such as adversarial model learning (Rigter et al., 2022), learning pessimistic models (Kidambi et al., 2020; Guo et al., 2022), using model ensembles to form penalties (Yu et al., 2020), or combining model and values (Yu et al., 2021).

The last category aims to learn conservative values such as fitted Q-iteration using conservative update (Liu et al., 2020), conservative Q-learning (CQL) (Kumar et al., 2020), critic regularization (Kostrikov et al., 2021), and subtracting penalties (Rezaeifar et al., 2022).

On the theoretical side, many works use some form of uncertainty quantification to design to ensure pessimism (Yin and Wang, 2021; Kumar et al., 2021; Uehara et al., 2021; Yin et al., 2022; Zhang et al., 2022; Yan et al., 2022; Shi and Chi, 2022; Wang et al., 2022). Except for uncertainty

quantification, in linear function approximation settings, Zanette et al. (2021) uses value function perturbation combined with the actor-critic method. Recent advances in the theory of pessimistic algorithms include MIS (Zhan et al., 2022; Chen and Jiang, 2022) and adversarially trained actor-critic (ATAC) (Cheng et al., 2022). In particular, our algorithm is built based on MIS combined with the actor-critic method.

# B   Proof of Proposition 2.1

The first part of the proposition is easy to see:

$$(C_{\ell_2}^\pi)^2 = \|w^\pi\|_{2,\mu}^2 = \mathbb{E}_\mu[(w^\pi(s,a))^2] \le \mathbb{E}_\mu\left[C_{\ell_\infty}^\pi \frac{d^\pi(s,a)}{\mu(s,a)}\right] = C_{\ell_\infty}^\pi \mathbb{E}_\mu\left[\frac{d^\pi(s,a)}{\mu(s,a)}\right] = C_{\ell_\infty}^\pi,$$

$$C_{\ell_2}^\pi = \|w^\pi\|_{2,\mu} \le \|w^\pi\|_\infty = C_\infty^\pi.$$

For the second part, consider the case where for a fixed policy $\pi$ and for any $s \in \mathcal{S}, a \in \mathcal{A}$, there exists $f \in \mathcal{F}$ such that $(f - \mathcal{T}^\pi f)(s,a)$ is non-zero only at $(s,a)$ and is zero otherwise. The Bellman-consistent concentrability defines $C_{\text{Bellman}}^\pi$ to be the smallest constant that

$$\max_{f \in \mathcal{F}} \frac{\|f - \mathcal{T}^\pi f\|_{2,d^\pi}^2}{\|f - \mathcal{T}^\pi f\|_{2,\mu}^2} \le C_{\text{Bellman}}^\pi$$

$$\implies \max_{s,a} \frac{(f - \mathcal{T}^\pi f)^2(s,a)d^\pi(s,a)}{(f - \mathcal{T}^\pi f)^2(s,a)\mu(s,a)} \le C_{\text{Bellman}}^\pi$$

$$\implies \max_{s,a} \frac{d^\pi(s,a)}{\mu(s,a)} \le C_{\text{Bellman}}^\pi,$$

which makes it equal to the $\ell_\infty$ variant. On the other hand, the $\ell_2$ variant only requires the average importance weights to be bounded:

$$\sum_{s,a} \left(\frac{d^\pi(s,a)}{\mu(s,a)}\right)^2 \mu(s,a) = \mathbb{E}_{d^\pi}\left[\frac{d^\pi(s,a)}{\mu(s,a)}\right] \le (C_{\ell_2}^\pi)^2.$$

# C   Theoretical Analysis of the Main Theorem

In this section, we provide theoretical proof of our main theorem (Theorem 1). We first present two key lemmas in Appendix C.1 and then prove the main theorem in Appendix C.3. For convenience, we always assume Assumptions 1 to 3 hold.

## C.1   Key lemmas

The first lemma shows that with high probability, the population version of our weighted average Bellman error regularizer is close to the empirical version.

**Lemma 2** (Concentration of the empirical regularizer). *With probability at least $1 - \delta$, for any $f \in \mathcal{F}$, $\pi \in \Pi$, we have*

$$|\mathcal{E}_\mu(\pi, f) - \mathcal{E}_\mathcal{D}(\pi, f)| \le \epsilon_{stat}.$$

*Proof.* We condition on the high probability event in Lemma 4. For any $f \in \mathcal{F}$ and $\pi \in \Pi$, define $w_{\pi,f}^* = \arg\max_{w \in \mathcal{W}} \mathcal{E}_\mu(\pi, f) = \arg\max_{w \in \mathcal{W}} |\mathbb{E}_\mu[w(s,a)(f - \mathcal{T}^\pi f)(s,a)]|$ and define $\hat{w}_{\pi,f} =$

$\arg\max_{w \in \mathcal{W}} \mathcal{E}_{\mathcal{D}}(\pi, f) = \arg\max_{w \in \mathcal{W}} \left| \frac{1}{N} \sum_{(s,a,r,s') \in \mathcal{D}} w(s,a)(f(s,a) - r - \gamma f(s', \pi)) \right|$. Then

$$\mathcal{E}_\mu(\pi, f) - \mathcal{E}_{\mathcal{D}}(\pi, f)$$

$$= |\mathbb{E}_\mu[w^*_{\pi,f}(s,a)(f - \mathcal{T}^\pi f)(s,a)]| - \left| \frac{1}{N} \sum_{(s,a,r,s') \in \mathcal{D}} \hat{w}_{\pi,f}(s,a)(f(s,a) - r - \gamma f(s', \pi)) \right|$$

$$= |\mathbb{E}_\mu[w^*_{\pi,f}(s,a)(f - \mathcal{T}^\pi f)(s,a)]| - |\mathbb{E}_\mu[\hat{w}_{\pi,f}(s,a)(f - \mathcal{T}^\pi f)(s,a)]|$$

$$+ |\mathbb{E}_\mu[\hat{w}_{\pi,f}(s,a)(f - \mathcal{T}^\pi f)(s,a)]| - \left| \frac{1}{N} \sum_{(s,a,r,s') \in \mathcal{D}} \hat{w}_{\pi,f}(s,a)(f(s,a) - r - \gamma f(s', \pi)) \right|$$

$$\geq 0 - \epsilon_{\text{stat}} = -\epsilon_{\text{stat}},$$

where the inequality holds by the optimality of $w^*_{\pi,f}$ and Lemma 4. Similarly,

$$\mathcal{E}_\mu(\pi, f) - \mathcal{E}_{\mathcal{D}}(\pi, f)$$

$$= |\mathbb{E}_\mu[w^*_{\pi,f}(s,a)(f - \mathcal{T}^\pi f)(s,a)]| - \left| \frac{1}{N} \sum_{(s,a,r,s') \in \mathcal{D}} w^*_{\pi,f}(s,a)(f(s,a) - r - \gamma f(s', \pi)) \right|$$

$$+ \left| \frac{1}{N} \sum_{(s,a,r,s') \in \mathcal{D}} w^*_{\pi,f}(s,a)(f(s,a) - r - \gamma f(s', \pi)) \right|$$

$$- \left| \frac{1}{N} \sum_{(s,a,r,s') \in \mathcal{D}} \hat{w}_{\pi,f}(s,a)(f(s,a) - r - \gamma f(s', \pi)) \right|$$

$$\leq \epsilon_{\text{stat}} + 0 = \epsilon_{\text{stat}},$$

where the inequality holds by the optimality of $\hat{w}_{\pi,f}$ and Lemma 4. $\qquad \square$

The next lemma provides a high-probability upper bound of the empirical weighted average Bellman error of $f_\pi$ w.r.t. $\pi$, where $f_\pi$ is the (approximate) $Q$-function of $\pi$.

**Lemma 3** (Empirical weighted average Bellman error of approximate $Q$ function)**.** *With probability at least $1 - \delta$, for any $\pi \in \Pi$, we have*

$$\mathcal{E}_{\mathcal{D}}(\pi, f_\pi) \leq C^\star_{\ell_2} \sqrt{\epsilon_{\mathcal{F}}} + \epsilon_{stat}.$$

*where $f_\pi = \arg\min_{f \in \mathcal{F}} \max_{admissible \, \nu} \|f - \mathcal{T}^\pi f\|^2_{2,\nu}$.*

*Proof.* We condition on the high probability event in Lemma 2. Since

$$\mathcal{E}_\mu(\pi, f_\pi) = \max_{w \in \mathcal{W}} |\mathbb{E}_\mu[w(s,a)(f_\pi - \mathcal{T}^\pi f_\pi)(s,a)]|$$

$$\leq \max_{w \in \mathcal{W}} \|w\|_{2,\mu} \|f_\pi - \mathcal{T}^\pi f_\pi\|_{2,\mu}$$

$$\leq C^\star_{\ell_2} \sqrt{\epsilon_{\mathcal{F}}},$$

where the first inequality is by Cauchy-Schwarz inequality and the second inequality is by the definition of $f_\pi$ and Assumption 1, we can immediately obtain that

$$\mathcal{E}_{\mathcal{D}}(\pi, f_\pi) \leq \mathcal{E}_\mu(\pi, f_\pi) + \epsilon_{\text{stat}} = C^\star_{\ell_2} \sqrt{\epsilon_{\mathcal{F}}} + \epsilon_{\text{stat}}.$$

$\qquad \square$

## C.2    Complementary lemmas

We provide two complementary lemmas in this section, which are both high-probability concentration inequalities.

**Lemma 4** (Concentration of weighted average Bellman error). *With probability at least $1 - \delta$, for any $f \in \mathcal{F}$, $\pi \in \Pi$ and $w \in \mathcal{W}$, we have*

$$\left| |\mathbb{E}_\mu[(f - \mathcal{T}^\pi f)w]| - \left| \frac{1}{N} \sum_{(s,a,r,s') \in \mathcal{D}} w(s,a)(f(s,a) - r - \gamma f(s', \pi)) \right| \right|$$

$$\leq O\left( V_{\max} C_{\ell_2}^\star \sqrt{\frac{\log(|\mathcal{F}||\Pi||\mathcal{W}|/\delta)}{N}} + \frac{V_{\max} B_w \log(|\mathcal{F}||\Pi||\mathcal{W}|/\delta)}{N} \right) \triangleq \epsilon_{stat}.$$

*Proof.* It suffices to bound

$$\left| \mathbb{E}_\mu[(f - \mathcal{T}^\pi f)w] - \frac{1}{N} \sum_{(s,a,r,s') \in \mathcal{D}} w(s,a)(f(s,a) - r - \gamma f(s', \pi)) \right|$$

for any fixed $f \in \mathcal{F}$, $\pi \in \Pi$, $w \in \mathcal{W}$, and a union bound and the triangle inequality conclude.

Note that

$$\mathbb{E}_\mu\left[ \frac{1}{N} \sum_{(s,a,r,s') \in \mathcal{D}} (f(s,a) - r - \gamma f(s', \pi))w(s,a) \right]$$

$$= \mathbb{E}_{(s,a) \sim \mu, r \sim r(s,a), s' \sim P(\cdot|s,a)} \left[ (f(s,a) - r - \gamma f(s', \pi))w(s,a) \right]$$

$$= \mathbb{E}_{(s,a) \sim \mu} \mathbb{E}_{r \sim r(s,a), s' \sim P(\cdot|s,a)} \left[ (f(s,a) - r - \gamma f(s', \pi))w(s,a)|s,a \right]$$

$$= \mathbb{E}_{(s,a) \sim \mu}[(f(s,a) - \mathcal{T}^\pi f(s,a))w(s,a)].$$

Then by Bernstein's inequality, we have that with probability at least $1 - \delta$,

$$\left| \mathbb{E}_\mu[(f - \mathcal{T}^\pi f)w] - \frac{1}{N} \sum_{(s,a,r,s') \in \mathcal{D}} (f(s,a) - r - \gamma f(s', \pi))w(s,a) \right|$$

$$\leq O\left( \sqrt{\frac{\mathsf{Var}_\mu[(f - \mathcal{T}^\pi f)w] \log(1/\delta)}{N}} + \frac{V_{\max} B_w \log(1/\delta)}{N} \right).$$

Since

$$\mathsf{Var}_\mu[(f - \mathcal{T}^\pi f)w] \leq \mathbb{E}_\mu[(f - \mathcal{T}^\pi f)^2 w^2] \leq O(V_{\max}^2 \|w\|_{2,\mu}^2) \leq O(V_{\max}^2 (C_{\ell_2}^\star)^2),$$

we can obtain that

$$\left| \mathbb{E}_\mu[(f - \mathcal{T}^\pi f)w] - \frac{1}{N} \sum_{(s,a,r,s') \in \mathcal{D}} (f(s,a) - r - \gamma f(s', \pi))w(s,a) \right|$$

$$\leq O\left( V_{\max} C_{\ell_2}^\star \sqrt{\frac{\log(1/\delta)}{N}} + \frac{V_{\max} B_w \log(1/\delta)}{N} \right),$$

which implies the result. $\qquad\square$

**Lemma 5** (Concentration of the actor's objective). *With probability at least $1 - \delta$, for any $f \in \mathcal{F}$, $\pi \in \Pi$, we have*

$$|\mathcal{L}_\mu(\pi, f) - \mathcal{L}_\mathcal{D}(\pi, f)| \leq O\left( V_{\max} \sqrt{\frac{\log(|\mathcal{F}||\Pi|/\delta)}{N}} \right) \leq \epsilon_{stat}$$

*where $\epsilon_{stat}$ is defined in Lemma 4.*

*Proof.* Note that $\mathbb{E}_\mu[\mathcal{L}_\mathcal{D}(\pi, f)] = \mathcal{L}_\mu(\pi, f)$ and $|f(s, \pi) - f(s, a)| \leq O(V_{\max})$. Applying a Hoeffding's inequality for any fixed $f$, $\pi$ and a union bound over all $f \in \mathcal{F}$, $\pi \in \Pi$, we can obtain that with probability at least $1 - \delta$, it holds that

$$|\mathcal{L}_\mu(\pi, f) - \mathcal{L}_\mathcal{D}(\pi, f)| \leq O\left( V_{\max} \sqrt{\frac{\log(|\mathcal{F}||\Pi|/\delta)}{N}} \right).$$

for all $f \in \mathcal{F}$ and $\pi \in \Pi$. $\qquad\square$

## C.3  Proof of Theorem 1

Now we are able to prove our main theorem equipped with lemmas in previous sections.

*Proof of Theorem 1.* By Lemma 1 and the definition of $\bar\pi$, we have

$$J(\pi) - J(\bar\pi) = \frac{1}{K}\sum_{k=1}^{K}(J(\pi) - J(\pi_k))$$

$$=\frac{1}{K}\sum_{k=1}^{K}(\underbrace{\mathbb{E}_\mu[(f_k - \mathcal{T}^{\pi_k}f_k)(s,a)]}_{(a)} + \underbrace{\mathbb{E}_\pi[(\mathcal{T}^{\pi_k}f_k - f_k)(s,a)]}_{(b)}$$

$$+ \underbrace{\mathbb{E}_\pi[f_k(s,\pi) - f_k(s,\pi_k)]}_{(c)} + \underbrace{\mathcal{L}_\mu(\pi_k, f_k) - \mathcal{L}_\mu(\pi_k, Q^{\pi_k})}_{(d)}).$$

Now we condition on the high probability event in Lemma 2 and Lemma 5 simultaneously and rescale $\delta$ by $1/2$ and apply a union bound. Note that $d^\pi/\mu \in \mathcal{W}$, which implies that $(a) + (b) \le 2\mathcal{E}_\mu(\pi_k, f_k) \le 2\mathcal{E}_\mathcal{D}(\pi_k, f_k) + 2\epsilon_{\text{stat}}$, where the last inequality holds by Lemma 2. By Lemma 13 of Cheng et al. (2022), we can obtain that

$$|\mathcal{L}_\mu(\pi_k, Q^{\pi_k}) - \mathcal{L}_\mu(\pi_k, f_{\pi_k})| \le \|f_{\pi_k} - \mathcal{T}^{\pi_k}f_{\pi_k}\|_{2,\mu} + \|f_{\pi_k} - \mathcal{T}^{\pi_k}f_{\pi_k}\|_{2,d^{\pi_k}} \le O(\sqrt{\epsilon_\mathcal{F}}),$$

where $f_{\pi_k} \triangleq \arg\min_{f\in\mathcal{F}}\max_{\text{admissible }\nu}\|f - \mathcal{T}^\pi f\|_{2,\nu}^2$. Also, by Lemma 5, we have

$$|\mathcal{L}_\mu(\pi_k, f_k) - \mathcal{L}_\mathcal{D}(\pi_k, f_k)| + |\mathcal{L}_\mu(\pi_k, f_{\pi_k}) - \mathcal{L}_\mathcal{D}(\pi_k, f_{\pi_k})| \le O(\epsilon_{\text{stat}}).$$

Therefore,

$$\begin{aligned}(a)+(b)+(d) \le & \mathcal{L}_\mu(\pi_k, f_k) + 2\mathcal{E}_\mathcal{D}(\pi_k, f_k) + 2\epsilon_{\text{stat}} - \mathcal{L}_\mu(\pi_k, f_{\pi_k}) + O(\sqrt{\epsilon_\mathcal{F}})\\ \le & \mathcal{L}_\mathcal{D}(\pi_k, f_k) + 2\mathcal{E}_\mathcal{D}(\pi_k, f_k) + O(\epsilon_{\text{stat}}) - \mathcal{L}_\mathcal{D}(\pi_k, f_{\pi_k}) + O(\sqrt{\epsilon_\mathcal{F}})\\ \le & \mathcal{L}_\mathcal{D}(\pi_k, f_{\pi_k}) + 2\mathcal{E}_\mathcal{D}(\pi_k, f_{\pi_k}) + O(\epsilon_{\text{stat}}) - \mathcal{L}_\mathcal{D}(\pi_k, f_{\pi_k}) + O(\sqrt{\epsilon_\mathcal{F}})\\ \le & O(\epsilon_{\text{stat}} + C_{\ell_2}^\star\sqrt{\epsilon_\mathcal{F}}).\end{aligned}$$

where the third inequality holds by the optimality of $f_k$, and the last inequality holds by Lemma 3. Therefore,

$$J(\pi) - J(\bar\pi) \le O(\epsilon_{\text{stat}} + C_{\ell_2}^\star\sqrt{\epsilon_\mathcal{F}}) + \epsilon_{\text{opt}}^\pi.$$

$\square$

# D  Analysis of Robust Policy Improvement

*Proof of Theorem 2.* By Lemma 1 and the definition of $\bar\pi$, we have

$$J(\mu) - J(\bar\pi) = \frac{1}{K}\sum_{k=1}^{K}(J(\mu) - J(\pi_k))$$

$$=\frac{1}{K}\sum_{k=1}^{K}(\underbrace{\mathbb{E}_\mu[(f_k - \mathcal{T}^{\pi_k}f_k)(s,a)]}_{(a)} + \underbrace{\mathbb{E}_\mu[(\mathcal{T}^{\pi_k}f_k - f_k)(s,a)]}_{(b)}$$

$$+ \underbrace{\mathbb{E}_\pi[f_k(s,\mu) - f_k(s,\pi_k)]}_{(c)} + \underbrace{\mathcal{L}_\mu(\pi_k, f_k) - \mathcal{L}_\mu(\pi_k, Q^{\pi_k})}_{(d)}).$$

Now we condition on the high probability event in Lemma 2 and Lemma 5 simultaneously and rescale $\delta$ by $1/2$ and apply a union bound. Note that $(a) + (b) = 0$. By Lemma 13 of Cheng et al. (2022), we can obtain that

$$|\mathcal{L}_\mu(\pi_k, Q^{\pi_k}) - \mathcal{L}_\mu(\pi_k, f_{\pi_k})| \le \|f_{\pi_k} - \mathcal{T}^{\pi_k}f_{\pi_k}\|_{2,\mu} + \|f_{\pi_k} - \mathcal{T}^{\pi_k}f_{\pi_k}\|_{2,d^{\pi_k}} \le O(\sqrt{\epsilon_\mathcal{F}}), \quad (9)$$

where $f_{\pi_k} \triangleq \arg\min_{f \in \mathcal{F}} \max_{\text{admissible } \nu} \|f - \mathcal{T}^\pi f\|_{2,\nu}^2$. Also, by Lemma 5, we have

$$|\mathcal{L}_\mu(\pi_k, f_k) - \mathcal{L}_\mathcal{D}(\pi_k, f_k)| + |\mathcal{L}_\mu(\pi_k, f_{\pi_k}) - \mathcal{L}_\mathcal{D}(\pi_k, f_{\pi_k})| \leq O(\epsilon_{\text{stat}}). \tag{10}$$

Therefore,

$$
\begin{aligned}
&\mathcal{L}_\mu(\pi_k, f_k) - \mathcal{L}_\mu(\pi_k, Q^{\pi_k}) \\
\leq& \mathcal{L}_\mu(\pi_k, f_k) + \beta \mathcal{E}_\mathcal{D}(\pi_k, f_k) - \mathcal{L}_\mu(\pi_k, Q^{\pi_k}) && (\mathcal{E}_\mathcal{D}(\cdot, \cdot) \geq 0) \\
\leq& \mathcal{L}_\mu(\pi_k, f_k) + \beta \mathcal{E}_\mathcal{D}(\pi_k, f_k) - \mathcal{L}_\mu(\pi_k, Q^{\pi_k}) - \beta \mathcal{E}_\mathcal{D}(\pi_k, f_{\pi_k}) \\
&+ \beta C_{\ell_2}^\star \sqrt{\epsilon_\mathcal{F}} + \beta \epsilon_{\text{stat}} && (\text{Lemma 3}) \\
\lesssim& \mathcal{L}_\mathcal{D}(\pi_k, f_k) + \beta \mathcal{E}_\mathcal{D}(\pi_k, f_k) - \mathcal{L}_\mathcal{D}(\pi_k, f_{\pi_k}) - \beta \mathcal{E}_\mathcal{D}(\pi_k, f_{\pi_k}) \\
&+ (\beta + 1)(\epsilon_{\text{stat}} + C_{\ell_2}^\star \sqrt{\epsilon_\mathcal{F}}) && ((9), (10)) \\
\leq& (\beta + 1)(\epsilon_{\text{stat}} + C_{\ell_2}^\star \sqrt{\epsilon_\mathcal{F}}). && (\text{Optimality of } f_k)
\end{aligned}
$$

Therefore,

$$J(\mu) - J(\bar{\pi}) \lesssim (\beta + 1)(\epsilon_{\text{stat}} + C_{\ell_2}^\star \sqrt{\epsilon_\mathcal{F}}) + \epsilon_{\text{opt}}^\pi.$$

$\square$

# E  Analysis of the Suboptimality of ATAC

In this section, we prove Proposition 4.1 in Appendix E.1. For convenience, we use $\text{Bern}(p)$ to denote a Bernoulli variable with parameter $p$, and use $\text{Bin}(n, p)$ to denote a binomial variable with parameters $n$ and $p$.

## E.1  Proof of Proposition 4.1

**Construction of a two-arm bandit example.**  Assume there are two arms $a_1, a_2$. Assume $\beta = \tilde{\Theta}(N^\alpha)$ where $\alpha > \frac{1}{2}$. Note that this is a more general case than $\beta = \Theta(N^{2/3})$ as stated in the proposition. Let $\Delta = \min\{\beta/N, 1/10\}$. Assume the reward of the first arm is deterministic, i.e., $r(a_1) = 1/2 + \Delta$, and $r(a_2) = \text{Bern}(1/2)$. Let $\Pi = \{\pi_1, \pi_2\}$, where $\pi_1(a_1) = 1, \pi_1(a_2) = 0$, and $\pi_2(a_1) = 0, \pi_2(a_2) = 1$. Also, let $\mathcal{F} = \{f_1, f_2\}$, where $f_1(a_1) = 1/2 + \Delta, f_1(a_2) = 1/2$ and $f_2(a_1) = 1/2 + \Delta, f_2(a_2) = 1/2 + 2\Delta$. Finally, the dataset distribution $\mu$ satisfies $\mu(a_2) = \frac{1}{N\Delta^2}$, $\mu(a_1) = 1 - \mu(a_2)$. Note that in bandit settings, for any policy $\pi$ and any function $f$, we have $Q^\pi = \mathcal{T}^\pi f = r$. Therefore, the example above satisfies both completeness and realizability assumptions.

**Proof of the suboptimality of ATAC.**  The ATAC algorithm is simplified to

$$\hat{\pi} \in \arg\max_{\pi \in \Pi} \mathbb{E}_\mu[f^\pi(\pi) - f^\pi(a)],$$

$$\text{s.t. } f^\pi \in \arg\min_{f \in \mathcal{F}} \mathbb{E}_\mu[f(\pi) - f(a)] + \beta \mathbb{E}_\mu[((f - r)(a))^2],$$

in bandit settings.

For convenience, we assume that $\mathbb{P}_\mu(a = a_1) = \mathbb{P}_\mathcal{D}(a = a_1)$. Note that by anti-concentration of the binomial distribution (Lemma 6), we have that with constant probability, it holds that $\hat{r}(a_2) \geq \frac{1}{2} + \frac{2}{\sqrt{N\mu(a_2)}} = \frac{1}{2} + 2\Delta$.

Conditioned on this event, we compute $f^{\pi_1}$ and $f^{\pi_2}$ separately. We have

$$
\begin{aligned}
\mathcal{L}_\mathcal{D}(\pi_1, f_1) =& \mathbb{E}_\mu[f_1(\pi_1) - f_1(a)] = f_1(\pi_1) - \mu(a_1) f_1(a_1) - \mu(a_2) f_1(a_2) \\
=& \mu(a_2)(f_1(a_1) - f_1(a_2)) = \frac{1}{N\Delta}, \\
\mathcal{L}_\mathcal{D}(\pi_1, f_2) =& \mathbb{E}_\mu[f_2(\pi_1) - f_2(a)] = f_2(\pi_1) - \mu(a_1) f_2(a_1) - \mu(a_2) f_2(a_2) \\
=& \mu(a_2)(f_2(a_1) - f_2(a_2)) = -\frac{1}{N\Delta},
\end{aligned}
$$

and

$$\beta\mathcal{E}_{\mathcal{D}}(\pi_1, f_1) = \beta\mathbb{E}_\mu[((f_1 - r)(a))^2] = \beta\mu(a_2)\left(\frac{1}{2}\right)^2 = \frac{\beta\mu(a_2)}{4},$$

$$\begin{aligned}
\beta\mathcal{E}_{\mathcal{D}}(\pi_1, f_2) =& \beta\mathbb{E}_\mu[((f_2 - r)(a))^2] \\
=& \beta\mu(a_2)\left(\hat{r}(a_2)\left(\frac{1}{2} - 2\Delta\right)^2 + (1 - \hat{r}(a_2))\left(\frac{1}{2} + 2\Delta\right)^2\right) \\
\leq& \beta\mu(a_2)\left(\left(\frac{1}{2} + 2\Delta\right)\left(\frac{1}{2} - 2\Delta\right)^2 + \left(\frac{1}{2} - 2\Delta\right)\left(\frac{1}{2} + 2\Delta\right)^2\right) \\
\leq& \beta\mu(a_2)\left(\left(\frac{1}{2}\right)^2 - (2\Delta)^2\right) \\
=& \frac{\beta\mu(a_2)}{4} - 4\frac{\beta}{N}.
\end{aligned}$$

Also, we have

$$\begin{aligned}
\mathcal{L}_{\mathcal{D}}(\pi_2, f_1) =& \mathbb{E}_\mu[f_1(\pi_2) - f_1(a)] = f_1(\pi_2) - \mu(a_1)f_1(a_1) - \mu(a_2)f_1(a_2) \\
=& \mu(a_1)(f_1(a_2) - f_1(a_1)) = -\left(1 - \frac{1}{N\Delta^2}\right)\Delta, \\
\mathcal{L}_{\mathcal{D}}(\pi_2, f_2) =& \mathbb{E}_\mu[f_2(\pi_2) - f_2(a)] = f_2(\pi_2) - \mu(a_1)f_2(a_1) - \mu(a_2)f_2(a_2) \\
=& \mu(a_1)(f_2(a_2) - f_2(a_1)) = \left(1 - \frac{1}{N\Delta^2}\right)\Delta,
\end{aligned}$$

and

$$\beta\mathcal{E}_\mu(\pi_2, f_1) = \beta\mathcal{E}_{\mathcal{D}}(\pi_1, f_1) = \frac{\beta\mu(a_2)}{4},$$

$$\beta\mathcal{E}_\mu(\pi_2, f_2) = \beta\mathcal{E}_{\mathcal{D}}(\pi_1, f_2) \leq \frac{\beta\mu(a_2)}{4} - 4\frac{\beta}{N}.$$

Since $\beta/N \geq \Delta$, we can obtain that $\mathcal{L}_{\mathcal{D}}(\pi_2, f_1) + \beta\mathcal{E}_{\mathcal{D}}(\pi_2, f_1) > \mathcal{L}_{\mathcal{D}}(\pi_2, f_2) + \beta\mathcal{E}_{\mathcal{D}}(\pi_2, f_2)$, which implies that $f^{\pi_2} = f_2$. Finally, note that

$$\mathcal{L}_{\mathcal{D}}(\pi_2, f^{\pi_2}) = \mathcal{L}_{\mathcal{D}}(\pi_2, f_2) > \frac{\Delta}{2} \geq \frac{1}{N\Delta} = |\mathcal{L}_{\mathcal{D}}(\pi_1, f^{\pi_1})|,$$

we have $\hat{\pi} = \pi_2$ by ATAC algorithm. Note that $J(\pi_1) - J(\hat{\pi}) = r(\pi_1) - r(\pi_2) = \Delta \gg \tilde{\Omega}(1/\sqrt{N})$. Therefore, for any $\alpha > 1/2$, there exists an instance s.t. ATAC cannot achieve the optimal rate $O(1/\sqrt{N})$. In particular, when $\beta = \Theta(N^{2/3})$, we have $J(\pi_1) - J(\hat{\pi}) = \Delta = \Omega(1/N^{1/3})$.

### E.2 Complementary lemma

**Lemma 6** (Anti-concentration of Binomial distribution, adapted from Proposition 7.3.2 of Matoušek and Vondrák (2001))**.** *Let $X \sim Bin(n, \frac{1}{2})$ be a binomial random variable with mean $\mu = \frac{n}{2}$. Then we have that for any $t \in [0, \frac{1}{8}]$ and universal constants $c_1, c_2$,*

$$\Pr(X \geq \mu + nt) \geq c_1 e^{-c_2 t^2 n}.$$

## F  Additional Experimental Details and Results

### F.1  More implementation details

Our implementation is nearly identical to ATAC, except that we use a different regularizer for the critic. More details can be found in Cheng et al. (2022).

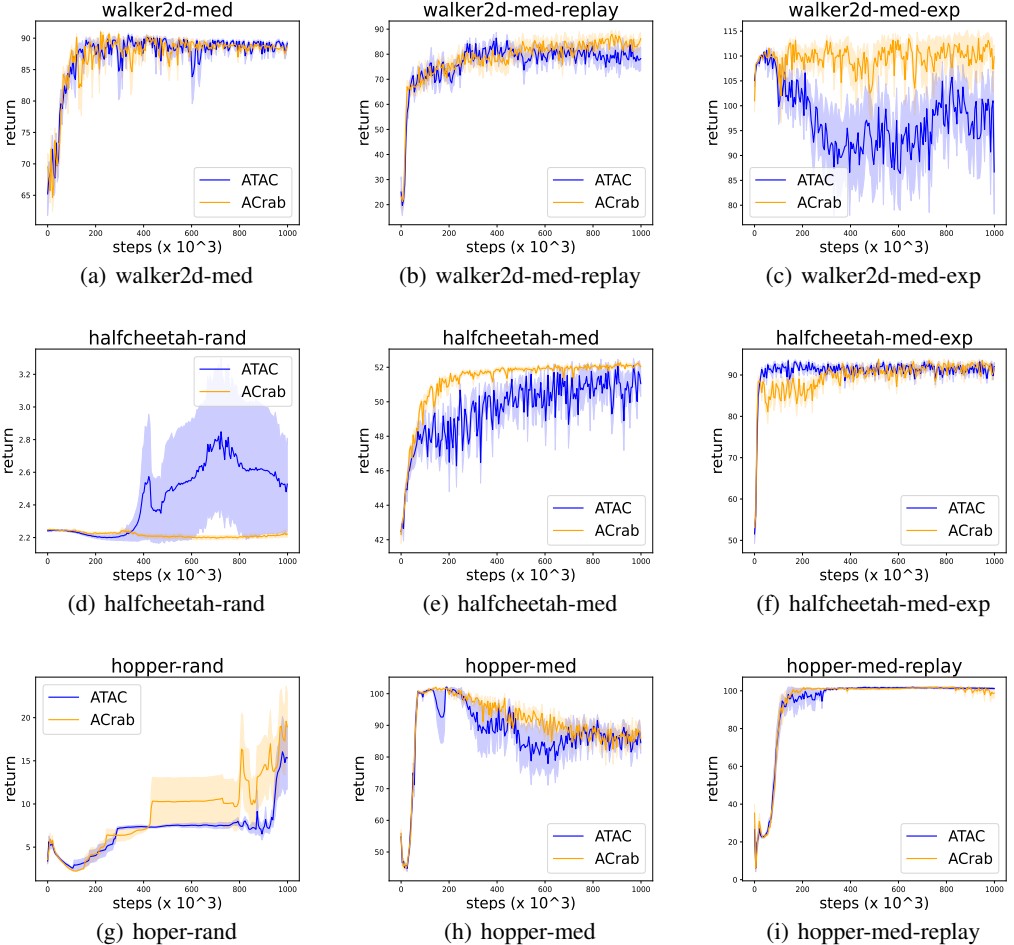

Figure 3: Comparison of A-Crab and ATAC in more different settings.

## F.2 Comparision of A-Crab and ATAC in different settings

In this section, we show comparisons of A-Crab and ATAC in more different settings in Figure 3.

## F.3 Value of $\beta$ and $C_\infty$ for each setting

In this section, we show our choices of $\beta$ and $C_\infty$ for each setting in Table F.3. Note that we directly choose the same value of $\beta$ as in ATAC, and select $C_\infty$ by a grid search over $\{1, 2, 5, 10, 20, 50, 100, 200\}$.

Table 2: Choices of $\beta$ and $C_\infty$ for each setting. The value of $\beta$ is the same as in ATAC (Cheng et al., 2022) and $C_\infty$ is chosen by a grid search.

|  | $\beta$ | $C_\infty$ |
|---|---|---|
| walker2d-random | 64 | 50 |
| walker2d-medium | 64 | 1 |
| walker2d-medium-replay | 64 | 5 |
| walker2d-medium-expert | 64 | 2 |
| hopper-random | 64 | 2 |
| hopper-medium | 64 | 1 |
| hopper-medium-replay | 16 | 1 |
| hopper-medium-expert | 1 | 1 |
| halfcheetah-random | 16 | 10 |
| halfcheetah-medium | 4 | 1 |
| halfcheetah-medium-replay | 16 | 2 |
| halfcheetah-medium-expert | 0.062 | 5 |

