# OpenReview forum: "Importance Weighted Actor-Critic for Optimal Conservative Offline Reinforcement Learning"
_NeurIPS.cc/2023/Conference — NeurIPS 2023 poster_

### Official Review · Reviewer_22cc · 2023-06-29

**Soundness:** 2 fair
**Presentation:** 3 good
**Contribution:** 2 fair
**Rating:** 6
**Confidence:** 3

**Summary:**

This paper introduces A-Crab, an offline RL algorithm derived from ATAC, which incorporates a modified loss function for the Q-function. Instead of employing the square Bellman error used in ATAC, A-Crab utilizes the importance-weighted Bellman error. With this modified loss function, A-Crab effectively addresses some disadvantages of ATAC. These include alleviating the visitation coverage assumption, enhancing the suboptimality rate, and simplifying the minimax optimization process. Additionally, A-Crab inherits several benefits from ATAC. Theoretical findings presented in this paper demonstrate the superior advantages of A-Crab in comparison to ATAC.

**Strengths:**

1. This paper is well-organized, presenting a table that compares provable offline RL algorithms, outlining the key distinctions of the proposed algorithm, A-Crab, in contrast to ATAC, and highlighting the advantages brought about by these modifications.
2. By simply incorporating the importance sampling ratio from the loss function of ATAC, A-Crab exhibits significantly improved characteristics, namely: 1) relaxation of the visitation coverage assumption, 2) enhancement of the suboptimal rate, and 3) simplification of the minimax problem into a maximization problem. These advancements result from non-trivial steps and can be accomplished by making only a few adjustments to the terms in the loss functions of ATAC.

**Weaknesses:**

1. Despite the noteworthy novelty of this paper in terms of presenting theoretical advantages of A-Crab, the absence of empirical results diminishes its impact. I acknowledge that it is common to solely provide theoretical results in the realm of provable offline RL. However, considering that A-Crab is built upon ATAC, which offers both theoretical and numerical results, it would be feasible to compare the performance of A-Crab with that of ATAC. I believe that including such results would significantly enhance the paper.
2. The importance sampling ratio in offline RL can become excessively large due to the limited coverage of the state-action space in the offline dataset and substantial differences between the current policy and the dataset policy. Consequently, the A-Crab algorithm may encounter instability when attempting to learn a robust policy in practical scenarios. However, if this paper addresses this issue, it would substantially strengthen its credibility.

**Questions:**

1. [Regarding Weakness 1] Would it be possible for you to present empirical results on D4RL datasets that compare the performance of A-Crab with that of ATAC?

**Limitations:**

The only limitation of this paper is the absence of empirical results. However, the implementation of A-Crab appears to be relatively straightforward, as it can be achieve by replacing the squared Bellman error in ATAC's implementation with the importance-weighted averaged Bellman error. Given that the implementation of ATAC is already available on GitHub, there is an opportunity to enhance this paper by presenting empirical results that compare the performance of A-Crab with that of ATAC.

---

> ### Author Rebuttal · Authors · 2023-08-09
>
> We thank the reviewer for the helpful and insightful comments. Below are our responses.
>
> >Would it be possible for you to present empirical results on D4RL datasets that compare the performance of A-Crab with that of ATAC?
>
> Yes. Please see the “global” response for details.
>
> >The importance sampling ratio in offline RL can become excessively large due to the limited coverage of the state-action space in the offline dataset and substantial differences between the current policy and the dataset policy. Consequently, the A-Crab algorithm may encounter instability when attempting to learn a robust policy in practical scenarios.
>
> Note that our algorithm only requires that the dataset covers the policy we aim to learn ( the goal policy), instead of covering all policies. In other words, we only require that the importance sampling ratio of the goal policy be bounded in a weighted $\ell_2$ sense. It does not matter whether the current policy is well covered by the dataset or has a bounded importance sampling ratio since our algorithm does not require computing the ratio for the current policy. Also, in our practical implementation, we can avoid using the $w$ function by using a computationally more efficient approximation of the importance-weighted Bellman error (see “global” response for details).

---

> > ### Comment · Reviewer_22cc · 2023-08-14
> >
> > Thank you for your kind responses to my questions. I am delighted to see the empirical results on D4RL tasks, comparing A-Crab with ATAC. These results can help readers better understand the effectiveness of the proposed method.

---

> > > ### Author Response · Authors · 2023-08-14
> > >
> > > We thank the reviewer again for the valuable feedback, and thanks for the suggestion of adding experimental results that would significantly enhance our paper. Since we have addressed the concerns pointed out during the official review, would you like to consider raising your score accordingly?

---

### Official Review · Reviewer_htrq · 2023-07-05

**Soundness:** 4 excellent
**Presentation:** 3 good
**Contribution:** 3 good
**Rating:** 6
**Confidence:** 3

**Summary:**

The paper proposes an offline reinforcement learning algorithm called Actor-Critic Regularized by Average Bellman error (A-Crab). A-Crab modifies the pessimistic offline RL framework by replacing the usual squared TD error with an importance sampled TD error. Due to the linearity of the importance sampled TD error, overestimation does not occur, allowing the removal of the correction term. A-Crab achieves the optimal suboptimality rate with weaker assumptions and is computationally efficient. An improvement over the behavior policy is also guaranteed under a wide range of hyperparameters.

**Strengths:**

### Originality
The authors introduced a novel technique using the importance sampled TD error instead of the squared TD error. Due to its linearity, the correction for overestimation is unnecessary, allowing better suboptimality bounds and weaker assumptions. The analysis also becomes very simple with the help of Cauchy–Schwarz inequality.
### Quality
I could not find any technical flaws in the arguments presented in the paper.
### Clarity
The paper is overall well-written and easy to understand.
### Significance
A-Crab achieves the optimal statistical rate of $1/\sqrt{N}$ with assumptions that are weaker compared to other provable offline RL algorithms. Also, using importance sampling instead of squaring can be useful in areas other than offline RL theory.

**Weaknesses:**

1. The authors unrealistically assume the action space, the policy space, the importance sampling weight function space, and the value function space to be finite.
2. Experimental verification of the performance bounds on a simple toy example would be interesting.

**Questions:**

Line 70 states that the A-Crab enjoys an optimal statistical rate of $1/\sqrt{N}$. Does this mean the optimality of $O(1/\sqrt{N})$ is theoretically proved? Or does this just mean that it is the best suboptimality rate to be discovered?

**Limitations:**

The authors do not address the limitations of their work. Assumptions on the finiteness of the action space, the policy space, the importance sampling weight function space, and the value function space can be viewed as limitations of this work. I believe this work does not have a potential negative societal impact.

---

> ### Author Rebuttal · Authors · 2023-08-09
>
> We thank the reviewer for the helpful and insightful comments. Below are our responses.
>
> >The authors unrealistically assume the action space, the policy space, the importance sampling weight function space, and the value function space to be finite.
>
>
> The finite cardinality assumption on all the above function classes was made only for convenience. Actually, we can instead replace the finite cardinality assumption with the bounded log covering-number assumption. The replacement is straightforward and is basically the same as Appendix B in [1].
>
> >Experimental verification of the performance bounds on a simple toy example would be interesting.
>
> We included experimental verification in the “global” response.
>
> >Line 70 states that the A-Crab enjoys an optimal statistical rate of $1/\sqrt{N}$. Does this mean the optimality of
> $O(1/\sqrt{N})$ is theoretically proved? Or does this just mean that it is the best suboptimality rate to be discovered?
>
> An $O(1/\sqrt{N})$ rate is the best one can hope in terms of $N$, and this is due to the intrinsic statistical error. For example, when we want to estimate the expectation of a random variable $X$ using $N$ i.i.d. samples $X_1, \ldots, X_N$, with constant probability, $|\bar X - \mathbb{E}[X]| \geq \Omega(1/\sqrt{N})$ where $\bar X = \frac{1}{N} \sum_{i=1}^N X_i$ is the empirical mean. This means the best rate of error one can hope is $O(1/\sqrt{N})$. Similarly, it can be proved that there is a lower bound $\Omega(1/\sqrt{N})$ of the suboptimality of the learned algorithm, which means $O(1/\sqrt{N})$ is optimal in terms of $N$.
>
> **References:**
>
> [1] Cheng, C. A., Xie, T., Jiang, N., & Agarwal, A. (2022, June). Adversarially trained actor critic for offline reinforcement learning. In International Conference on Machine Learning (pp. 3852-3878). PMLR.

---

> > ### Comment · Reviewer_htrq · 2023-08-13
> >
> > Thank you for the rebuttal. It would be nice to have these explanations included in the final version of the paper.

---

> > > ### Author Response · Authors · 2023-08-13
> > >
> > > Thanks for your time reviewing our paper and reading our response. We will include the points you mentioned in our revision.

---

> > > ### Author Response · Authors · 2023-08-14
> > >
> > > We thank the reviewer again for their great efforts in reviewing our paper. Since we have addressed your concern on both theoretical and empirical sides (the two points in the weakness section), would you like to consider raising your score accordingly?

---

### Official Review · Reviewer_PQdU · 2023-07-06

**Soundness:** 3 good
**Presentation:** 3 good
**Contribution:** 3 good
**Rating:** 5
**Confidence:** 2

**Summary:**

The paper introduces A-Crab, which combines marginalized importance sampling with the actor-critic paradigm to achieve optimal statistical rate in offline RL. Fm theoretical analysis, this algorithm is also more computationally efficient and relies on a weaker average notion of policy coverage compared to prior work.

**Strengths:**

1. The paper is well-written, effectively summarizing prior work and building upon it to propose the new algorithm. The notations are clear and consistent.
2. The algorithm itself exhibits strong theoretical properties, such as optimal statistical rate and efficiency, making it promising for offline RL problems.



**Weaknesses:**

1. The empirical evaluation of the algorithm is a major concern. Since the theory only requires general function approximators, the gap between theory and practice may not be substantial. Experimental results help demonstrate the algorithm's effectiveness on computation cost and learning speed.

**Questions:**

1. In Line 224, where does the overestimation come from? Please provide clarification.
2. In Line 248, why does ATAC need to optimize two functions, f and g? The previous mention in Line 214 only refers to f.
3. In Line 280, how does the theorem also demonstrate robustness against model mismatch? Please elaborate on this point.

**Limitations:**

Please refer to the weaknesses section.

---

> ### Author Rebuttal · Authors · 2023-08-09
>
> We thank the reviewer for the helpful and insightful comments. Below are our responses.
>
> >The empirical evaluation of the algorithm is a major concern.
>
> We provided empirical results to demonstrate the algorithm’s effectiveness. See the “global” response for details.
>
> >In Line 224, where does the overestimation come from?
>
> The overestimation is caused when directly using the empirical version $\mathbb{E}\_{\mathcal{D}} \left[(f(s,a)-r - \gamma f(s', \pi))^2\right]$ to estimate the term $\mathbb{E}\_\mu [((f - \mathcal{T}^\pi f)(s,a))^2]$. To better illustrate, we use a simpler example. Assume there are two random variables $X$, $Y$, and we want to estimate
> $\mathbb{E}[ ( X - \mathbb{E}[Y|X])^2]$
> (note that $\mathcal{T}^\pi f(s,a)$ is also an expectation conditioned on $f(s,a)$). If we have $n$ samples of $(X\_i, Y\_i)$ pairs, and directly use $\frac{1}{n}\sum\_{i=1}^n (X\_i - Y\_i)^2$ as an empirical estimator, then it is an overestimation of $\mathbb{E}[(X - \mathbb{E}[Y|X])^2]$ since $\mathbb{E}[\frac{1}{n}\sum\_{i=1}^n (X\_i - Y\_i)^2] = \mathbb{E}[(X-Y)^2] \geq \mathbb{E}[(X-\mathbb{E}[Y|X])^2]$. This is essentially the same as the overestimation in Line 224.
>
> >In Line 248, why does ATAC need to optimize two functions, f and g?
>
> This is highly related to the previous question. The reason ATAC needs another function $g$ is to address the overestimation issue. As we already discussed, $\mathbb{E}\_{\mathcal{D}} \left[(f(s,a)-r - \gamma f(s', \pi))^2\right]$ is an overestimate of $\mathbb{E}\_\mu [((f - \mathcal{T}^\pi f)(s,a))^2]$, and the amount of the overestimation is roughly equal to $\min\_{g \in \mathcal{F}} \mathbb{E}\_{\mathcal{D}} \left[(g(s,a)-r - \gamma f(s', \pi))^2\right]$. Therefore, one should use  $\mathbb{E}\_{\mathcal{D}} \left[(f(s,a)-r - \gamma f(s', \pi))^2\right] - \min\_{g \in \mathcal{F}} \mathbb{E}\_{\mathcal{D}} \left[(g(s,a)-r - \gamma f(s', \pi))^2\right]$ as an unbiased estimator as also mentioned in Line 225 and 226.
>
> >In Line 280, how does the theorem also demonstrate robustness against model mismatch?
>
> In Theorem 1, the upper bound of the suboptimality contains a term $C^\star_{\ell_2} \sqrt{\epsilon_\mathcal{F}}$ where $\epsilon_\mathcal{F}$ quantifies the model mismatch, which is defined in Assumption 1 (Line 137). Note that when $\epsilon_\mathcal{F} = 0$, there is no model mismatch; when $\epsilon_\mathcal{F} > 0$, the additional suboptimality caused by model mismatch is at most $C^\star_{\ell_2} \sqrt{\epsilon_\mathcal{F}}$, which means that our algorithm is robust against model mismatch.

---

> > ### Comment · Reviewer_PQdU · 2023-08-14
> >
> > Thanks for the replies and additional experiments. My questions on the theoretical side have been solved. However, I still have concerns regarding the empirical results and the connections to the theory. For example, how does the performance difference change during training, and how is the bound influenced by the usage of complex function approximation? I hope the authors could add more empirical details in the revised version.

---

> > > ### Author Response · Authors · 2023-08-15
> > >
> > > We thank the reviewer for the time reviewing our paper, reading our response, and providing valuable feedback! We are glad that we have successfully addressed the reviewer's questions on the theoretical side.
> > >
> > > For the empirical results, the primary purpose is to (empirically) prove that our algorithm is practical and can achieve great performance. Compared to ATAC, our A-Crab algorithm achieves better or comparable performance in various settings, as shown in the plots in the "global response". This is consistent with our theoretical results that our algorithm has better sample complexity than the previous algorithm.
> > >
> > > Below we also respond to two specific questions the reviewer mentioned.
> > >
> > >  >how does the performance difference change during training
> > >
> > > The change of performance difference during training is indicated by the change of performance during the training as shown in our plots. However, since it is hard to know the performance of the optimal policy in those relatively complex environments, it is also hard to plot the performance difference directly. However, note that the performance difference can be viewed as a constant minus the performance in any specific environment, so the performance difference curve can be obtained by mirroring the performance curve along the horizontal axis up to a constant shift.
> > >
> > > >how is the bound influenced by the usage of complex function approximation
> > >
> > > Typically there is a tradeoff between the cardinality (or covering number) of the function class and the model misspecification. The richer the function class, the smaller the model-misspecification parameter. Roughly speaking, there should be an "optimal" size of the function class that best balances the cardinality and model misspecification. The function class complexity can be changed by using different architectures of neural networks. Since the main focus of our experiments conducted during the rebuttal session is to compare with the previous ATAC algorithm and show our algorithm is practical, we use the same architecture of networks as ATAC for fairness. Also, exploring the influence on the performance of different network architectures is not the main focus of this work, but it would be interesting to see whether a different network can achieve better results.

---

> > > > ### Comment · Reviewer_PQdU · 2023-08-20
> > > >
> > > > Thanks to the authors for the careful explanations. I will maintain the score and believe further experiments will benefit the paper.

---

> > > > > ### Author Response · Authors · 2023-08-21
> > > > >
> > > > > Thanks for your suggestions and discussions! We are available if you have any further questions.

---

### Official Review · Reviewer_W5Wf · 2023-07-07

**Soundness:** 3 good
**Presentation:** 3 good
**Contribution:** 3 good
**Rating:** 6
**Confidence:** 3

**Summary:**

This paper proposes a novel algorithm called A-Crab (Actor-Critic Regularized by Average Bellman Error) for offline reinforcement learning (RL) in complex environments with insufficient data coverage. The algorithm combines the marginalized importance sampling framework with the actor-critic paradigm and addresses the challenges of handling high-dimensional observations and minimal data coverage. The paper presents sufficient theoretical analysis to demonstrate the advantages of the proposed algorithm over existing methods.

**Strengths:**

- The paper proposes a novel algorithm that combines various techniques to address the challenges of offline RL in complex environments. The use of marginalized importance sampling and the average Bellman error regularization in the critic's objective are innovative and practical.
- The theoretical analysis provides insights into the statistical properties of the proposed algorithm, with a focus on achieving the optimal statistical rate in converging to the best policy covered in the offline dataset.
- The paper provides a comparison with existing provable offline RL algorithms, highlighting the strengths and advantages of the proposed A-Crab algorithm.


**Weaknesses:**

- The paper provides no empirical evaluations or demonstrations of the proposed algorithm. Neither does it shed light on the design of practical algorithms.
- There remain some issues unsolved in the paper. See the questions for details.
- Some related works [1,2] use a similar optimization problem as Eq. (2) and also focus on computing an prioritization weight $w$ for Bellman error. They should be cited and discussed.


**Questions:**

- There seems to be a lack of comparison between $C_{\text{bellman}}$ and $C_{l_2}$ : How to show $C_{l_2}$ is a weaker assumption than $C_{\text{bellman}}$?
- How can the A-Crab algorithm shed light on the design of practical algorithms?

**Limitations:**

References

[1] Kumar, Aviral, Abhishek Gupta, and Sergey Levine. "Discor: Corrective feedback in reinforcement learning via distribution correction." Advances in Neural Information Processing Systems 33 (2020): 18560-18572.

[2] Liu, Xu-Hui, et al. "Regret minimization experience replay in off-policy reinforcement learning." Advances in Neural Information Processing Systems 34 (2021): 17604-17615.

---

> ### Author Rebuttal · Authors · 2023-08-09
>
> We thank the reviewer for the helpful and insightful comments. Below are our responses.
>
> >The paper provides no empirical evaluations or demonstrations of the proposed algorithm. Neither does it shed light on the design of practical algorithms.
>
> We showed empirical evaluation results to demonstrate that our A-Crab algorithm is practical. See the “global” response for details.
>
> >There seems to be a lack of comparison between $C_{bellman}$ and $C_{\ell_2}$:  How to show $C_{\ell_2}$ is a weaker assumption than $C_{bellman}$?
>
> In Figure 1, page 3 of the paper, we provided an intuitive comparison of $C_{bellman}$ and $C_{\ell_2}$, where for any fixed policy $\pi$, $C_{\ell_2}$ remains unchanged while $C_{bellman}$ goes larger as the function class $\mathcal{F}$ gets richer. When $\mathcal{F}$ is extremely expressive, $C_{bellman}$ can be as large as $C_{\infty}$ and thus can be much larger than $C_{\ell_2}$. For an arbitrary $\mathcal{F}$, it is hard to compare $C_{bellman}$ and $C_{\ell_2}$. However, they are both smaller than the previously widely-used coverage notion $C_{\ell_\infty}$.
>
> >Some related works [1,2] use a similar optimization problem as Eq. (2) and also focus on computing an prioritization weight $w$ for Bellman error. They should be cited and discussed.
>
> Thanks for pointing out. We will cite these two related works and add a discussion.

---

> > ### Comment · Reviewer_W5Wf · 2023-08-19
> > **Response**
> >
> > I acknowledge the authors' rebuttal and I remain my rating towards acceptance.

---

> > > ### Author Response · Authors · 2023-08-21
> > >
> > > Thanks for your time providing valuable feedback and reading our response. We will incorporate your suggestions in the revision.

---

### Author Rebuttal · Authors · 2023-08-09

We thank all the reviewers for their helpful and insightful comments. Below we first address common issues. Since all the reviewers mentioned that adding experimental results would make our theoretical results more solid and significantly enhance our paper, we compared our A-Crab algorithm to the previous ATAC algorithm on 12 Mujoco datasets (v2) from D4RL offline RL benchmark (the same as the datasets used in the ATAC paper for their main results). The attached pdf file contains plots showing the performance of ACrab and ATAC during training. Each curve is averaged over 8 random seeds (we choose 0-7 as random seeds), and each training step corresponds to a batch size of 256 (10^6 steps is roughly 1000 epochs). It shows that our A-Crab has higher returns and smaller variance (which indicates a more stable training procedure) than ATAC in various environments and has at least comparable results to ATAC for almost all environments.

Now we discuss some details of the implementation of A-Crab. Since it is straightforward to implement our algorithm based on ATAC, and nearly all the hyperparameters we used are the same as ATAC, we mainly emphasize the difference between A-Crab and ATAC in implementation.

Note that we only need to replace the squared Bellman error regularizer for the critic in ATAC with our proposed weighted average Bellman error regularizer. Recall the definition of our proposed weighted average Bellman error regularizer
   $\mathcal{E}\_\mathcal{D}(\pi, f) = \max\_{w \in \mathcal{W}} \left| \mathbb{E}\_\mathcal{D}[w(s,a)(f(s,a)-r-\gamma f(s',\pi))]\right|. $
Since the calculation of $\mathcal{E}\_\mathcal{D}(\pi, f)$ requires solving an optimization problem w.r.t. importance weights $w$, for computational efficiency, we choose $\mathcal{W} = [0, C\_\infty]^{\mathcal{S} \times \mathcal{A}}$, and thus

 $ \mathcal{E}\_{\mathcal{D}}^{\text{app}}(\pi, f) = C\_\infty\max$ { $  \mathbb{E}\_{\mathcal{D}}[(f(s,a)-r-\gamma f(s',\pi))\_+],  \mathbb{E}\_{\mathcal{D}}[(r+\gamma f(s',\pi)-f(s,a))\_+] $ },

where $(\cdot)\_+ = \max\$ { $\cdot, 0$ } and $C_\infty$ can be viewed as a  hyperparameter. We also observed that using a combination of squared Bellman error and our average Bellman error achieves better performance in practice, and we conjecture the reason is that the squared Bellman error regularizer is computationally more efficient and statistically suboptimal, while our average Bellman error regularizer is statistically optimal while computationally less efficient, and thus the combination of these two regularizers can benefit the training procedure. Specifically, in implementation, we choose
   $ \frac{1}{2}\left(2\mathcal{E}\_\mathcal{D}^{\text{app}}(\pi, f)  + \beta \mathbb{E}\_\mathcal{D} [((f - \mathcal{T}^\pi f)(s,a))^2] \right)$
as the regularizer, while ATAC uses  $\beta \mathbb{E}\_\mathcal{D} [((f - \mathcal{T}^\pi f)(s,a))^2]$.
All hyperparameters are the same as ATAC, including $\beta$. For the additional hyperparameter $C\_\infty$, we do a grid search on { $1, 2, 5, 10, 20, 50, 100, 200$ }.

---

### Comment · Area_Chair_P3wh · 2023-08-19

To Authors:

Thank you for submitting your rebuttal. We appreciate your efforts in addressing the reviewers' comments and providing clarification.

To Reviewers:

We kindly request you to complete your response to the authors' rebuttal as soon as possible. Time is of the essence, and the deadline is fast approaching. Your timely feedback and expertise are crucial in ensuring a fair and thorough evaluation process. Please prioritize reviewing the authors' response and provide your final feedback accordingly.

---

### Decision · Program_Chairs · 2023-09-21

**Decision:**

Accept (poster)

**Comment:**

Weighting the pros and cons, overall, the reviewers believe this paper should be accepted.